# Akt phosphorylates insulin receptor substrate to limit PI3K-mediated PIP3 synthesis

**Alison L Kearney[1†], Dougall M Norris[1,2†‡], Milad Ghomlaghi[3,4†], Martin Kin Lok Wong[1], Sean J Humphrey[1], Luke Carroll[1], Guang Yang[1], Kristen C Cooke[1], Pengyi Yang[5,6], Thomas A Geddes[1,6], Sungyoung Shin[3,4], Daniel J Fazakerley[2], Lan K Nguyen[3,4]\*, David E James[1,7]\*, James G Burchfield[1]\***

[1]Charles Perkins Centre, School of Life and Environmental Sciences, University of Sydney, Sydney, Australia; [2]Metabolic Research Laboratories, Wellcome Trust-Medical Research Council Institute of Metabolic Science, University of Cambridge, Cambridge, United Kingdom; [3]Department of Biochemistry and Molecular Biology, School of Biomedical Sciences, Monash University, Clayton, Australia; [4]Biomedicine Discovery Institute, Monash University, Clayton, Australia; [5]Charles Perkins Centre, School of Mathematics and Statistics, University of Sydney, Sydney, Australia; [6]Computational Systems Biology Group, Children's Medical Research Institute, University of Sydney, Westmead, Australia; [7]School of Medical Sciences, University of Sydney, Sydney, Australia

**\*For correspondence:**
Lan.K.Nguyen@monash.edu
(LKN);
david.james@sydney.edu.au (DEJ);
james.burchfield@sydney.edu.au
(JGB)

†These authors contributed equally to this work

**Present address:** ‡Metabolic Research Laboratories, Wellcome-Medical Research Council Institute of Metabolic Science, University of Cambridge, Cambridge, United Kingdom

**Abstract** The phosphoinositide 3-kinase (PI3K)-Akt network is tightly controlled by feedback mechanisms that regulate signal flow and ensure signal fidelity. A rapid overshoot in insulin-stimulated recruitment of Akt to the plasma membrane has previously been reported, which is indicative of negative feedback operating on acute timescales. Here, we show that Akt itself engages this negative feedback by phosphorylating insulin receptor substrate (IRS) 1 and 2 on a number of residues. Phosphorylation results in the depletion of plasma membrane-localised IRS1/2, reducing the pool available for interaction with the insulin receptor. Together these events limit plasma membrane-associated PI3K and phosphatidylinositol (3,4,5)-trisphosphate (PIP3) synthesis. We identified two Akt-dependent phosphorylation sites in IRS2 at S306 (S303 in mouse) and S577 (S573 in mouse) that are key drivers of this negative feedback. These findings establish a novel mechanism by which the kinase Akt acutely controls PIP3 abundance, through post-translational modification of the IRS scaffold.

## Introduction

Signalling networks enable cells to respond to diverse environmental challenges and maintain cellular homeostasis (*Humphrey et al., 2015b*; *Kholodenko et al., 2012*; *Ubersax and Ferrell, 2007*). They comprise an array of regulatory mechanisms that ensure the most appropriate response is achieved. For example, positive feedback loops facilitate adequate signal amplification, while negative feedback loops enable rapid, tightly regulated responses to stimuli and prevent pathway hyperactivation (*Cheong and Levchenko, 2010*; *Tyson et al., 2003*). Negative feedback also ensures that signalling pathways are resistant to external perturbations (*Stelling et al., 2004*) and may optimise the energetic cost of signal transduction (*Anders et al., 2020*). Identifying and characterising feedback mechanisms in signal transduction pathways is pivotal since dysfunctional cell signalling

**eLife digest** For the body to work properly, cells must constantly 'talk' to each other using signalling molecules. Receiving a chemical signal triggers a series of molecular events in a cell, a so-called 'signal transduction pathway' that connects a signal with a precise outcome.

Disturbing cell signalling can trigger disease, and strict control mechanisms are therefore in place to ensure that communication does not break down or become erratic. For instance, just as a thermostat turns off the heater once the right temperature is reached, negative feedback mechanisms in cells switch off signal transduction pathways when the desired outcome has been achieved.

The hormone insulin is a signal for growth that increases in the body following a meal to promote the storage of excess blood glucose (sugar) in muscle and fat cells. The hormone binds to insulin receptors at the cell surface and switches on a signal transduction pathway that makes the cell take up glucose from the bloodstream.

If the signal is not engaged diseases such as diabetes develop. Conversely, if the signal cannot be adequately switched of cancer can develop. Determining exactly how insulin works would help to understand these diseases better and to develop new treatments. Kearney et al. therefore set out to examine the biochemical 'fail-safes' that control insulin signalling.

Experiments using computer simulations of the insulin signalling pathway revealed a potential new mechanism for negative feedback, which centred on a molecule known as Akt. The models predicted that if the negative feedback were removed, then Akt would become hyperactive and accumulate at the cell's surface after stimulation with insulin. Further manipulation of the 'virtual' insulin signalling pathway and studies of live cells in culture confirmed that this was indeed the case. The cell biology experiments also showed how Akt, once at the cell surface, was able to engage the negative feedback and shut down further insulin signalling. Akt did this by inactivating a protein required to pass the signal from the insulin receptor to the rest of the cell. Overall, this work helps to understand cell communication by revealing a previously unknown, and critical component of the insulin signalling pathway.

frequently underlies complex diseases (*Fazakerley et al., 2019*; *Mora-Garcia and Sakamoto, 1999*; *Nguyen and Kholodenko, 2016*).

The phosphoinositide 3-kinase (PI3K)-Akt signalling network is activated by several cellular receptors and plays a central role in regulating cell growth and metabolism (*Fruman et al., 2017*). Engagement of this pathway is a multi-step process involving the actions of lipid and protein kinases and phosphatases, and relies on modulation of protein-protein and protein-lipid interactions. For example, binding of insulin to the insulin receptor (IR) leads to receptor auto-phosphorylation, and subsequent binding of the insulin receptor substrate (IRS) 1 and 2 adaptors via their phosphotyrosine-binding (PTB) domains (*Boucher et al., 2014*). IRS proteins are then tyrosine-phosphorylated and bind the Src homology 2 (SH2) domain of PI3K p85, recruiting PI3K (consisting of p85 and p110 subunits) to the plasma membrane (PM) (*Mosthaf et al., 1996*; *Songyang et al., 1993*). PI3K phosphorylates phosphatidylinositol (4,5) bisphosphate at the PM to form phosphatidylinositol (3,4,5) trisphosphate (PIP3). Phosphatase and tensin homolog (PTEN) catalyses the reverse reaction (*Kabuyama et al., 1996*; *Lee et al., 2018*; *Maehama and Dixon, 1998*; *Myers et al., 1998*). PIP3 is a bioactive lipid that recruits effectors such as phosphoinositide-dependent protein kinase 1 (PDPK1) and Akt via their pleckstrin homology (PH) domains, leading to their accumulation at the PM (*Currie et al., 1999*). Here, Akt is phosphorylated at T309 (in Akt2; T308 in Akt1, T305 in Akt3) by PDPK1 and S474 (in Akt2; S473 in Akt1, S472 in Akt3) by mammalian target of rapamycin complex 2 (mTORC2) (*Alessi et al., 1996a*; *Sarbassov et al., 2005*). This leads to full activation of Akt kinase activity (*Kearney et al., 2019*), and substrate phosphorylation ensues. More than 100 substrates of Akt have been reported, which regulate a myriad of biological pathways (*Manning and Toker, 2017*). Importantly, dysregulated PI3K/Akt activity is linked to several diseases including cancer and type 2 diabetes (*Fruman et al., 2017*), making tight regulation essential.

Insulin-stimulated recruitment of Akt to the PM displays overshoot behaviour (a response where the initial maxima exceeds the final steady state) and oscillations (*Ebner et al., 2017*; *Norris et al.,*

*2017*). These dynamics are indicative of acute negative feedback signals that regulate Akt activation (*Behar and Hoffmann, 2010*; *Cheong and Levchenko, 2010*; *Nyman et al., 2012*). mTORC1 and S6-kinase (S6K) are activated downstream of Akt (*Manning and Toker, 2017*), and have been reported to put the 'brakes' on PI3K/Akt signalling through a variety of mechanisms. For example, mTORC1/S6K phosphorylates the mTORC2 component RICTOR to attenuate mTORC2 activity and hence Akt activity (*Dibble et al., 2009*; *Julien et al., 2010*). Furthermore, mTORC1/S6K-mediated phosphorylation of IRS1 limits PI3K/Akt activation (*Copps and White, 2012*; *Hançer et al., 2014*; *Harrington et al., 2004*; *Shah et al., 2004*; *Shah and Hunter, 2006*; *Tzatsos, 2009*; *Tzatsos and Kandror, 2006*; *Yoneyama et al., 2018*). However, it remains unclear whether these feedbacks are responsible for the overshoot and oscillatory behaviour in Akt activation. As PI3K, Akt and mTOR are targets for cancer therapeutics (*Porta et al., 2014*), a deep understanding of these network topologies is important to elucidate the consequences of pathway manipulation and ultimately provide opportunities for improving drug efficacy.

Here, we applied live cell imaging, biochemical assays, and iterative computational modelling to interrogate the feedback signals regulating Akt activation. Our data uncover a potent Akt-dependent, mTORC1-independent feedback mechanism. Upon activation, Akt depletes PM localised IRS1/2 to reduce its interaction with the IR. This limits PM-associated PI3K and PIP3 synthesis, constituting a strong negative feedback loop. This feedback is driven by changes in the phosphorylation of IRS1/2 on a spectrum of residues. Specifically, we identify IRS2 as a novel substrate of Akt and show that Akt-mediated phosphorylation of IRS2 at S306 (S303 in mouse) and S577 (S573 in mouse) are critical drivers of this feedback.

## Results

### A mechanistic model of proximal insulin signalling

The recruitment behaviour of Akt in response to insulin is indicative of negative feedback, however the source of this feedback is unknown. Given the complexity of feedback architectures, we developed a computational model of insulin signalling to help dissect potential origins of negative feedback in this system (*Figure 1A*). Our model included nodes representing proximal insulin signalling (IR, IRS/PI3K) and accounted for various intricacies of Akt activation, such as its phosphorylation states (at both T309 and S474) as well as the positive feedback loop onto mTORC2 in response to SIN1 T86 phosphorylation (*Humphrey et al., 2013*; *Yang et al., 2015*). The model also included the previously described negative feedback loop from mTORC1/S6K to IRS/PI3K. As mTORC1 activation requires Akt-mediated phosphorylation of AKT1S1/PRAS40 (*Sancak et al., 2007*; *Wang et al., 2007*), we incorporated this feedback mechanism into the model using phospho-PRAS40 (T246) as a surrogate for mTORC1 activation (*Figure 1A*). The new model was formulated using ordinary differential equations (ODEs) and implemented in MATLAB that mathematically represents the network interactions as a series of ODEs based on established kinetic laws (see Materials and methods and *Supplementary file 1* for detailed model descriptions).

To train and evaluate our model, we first assessed the recruitment of TagRFP-T tagged Akt2 (described previously *Norris et al., 2017*) in live 3T3-L1 adipocytes using total internal reflection fluorescence microscopy (TIRFM). This cell line was used as they are exquisitely insulin responsive, providing a high signal-to-noise ratio for evaluating phenotypic responses. In response to 1 nM insulin, plasma membrane (PM)-associated Akt displayed overshoot behaviour. Specifically, PM levels of Akt reached a maximum at 75 s post-insulin stimulation, followed by a rapid decline that reached a new steady state that was roughly half of its maximum by 10 min post-stimulation (*Figure 1B*). In response to 100 nM insulin, the maximum recruitment of Akt was fourfold higher than 1 nM insulin, with an initial overshoot followed by a secondary increase (*Figure 1B*), which may reflect the engagement of a positive feedback signal at this dose. A similar overshoot was observed for insulin-stimulated phosphorylation of Akt at its activating sites T309 and S474 (*Figure 1C,D*), indicating that the overshoot in Akt recruitment is reflected in its activation. The kinetics of T309 phosphorylation were much faster than S474 phosphorylation. Interestingly the phosphorylation of Akt substrates (AS160, FOXO1, GSK3, and PRAS40) did not exhibit an overshoot (*Figure 1C,D*). Although Akt singly phosphorylated at T309 is active, the delayed onset of S474 phosphorylation doubles Akt activity (*Kearney et al., 2019*) and likely extends the peak phosphorylation observed for these substrates.

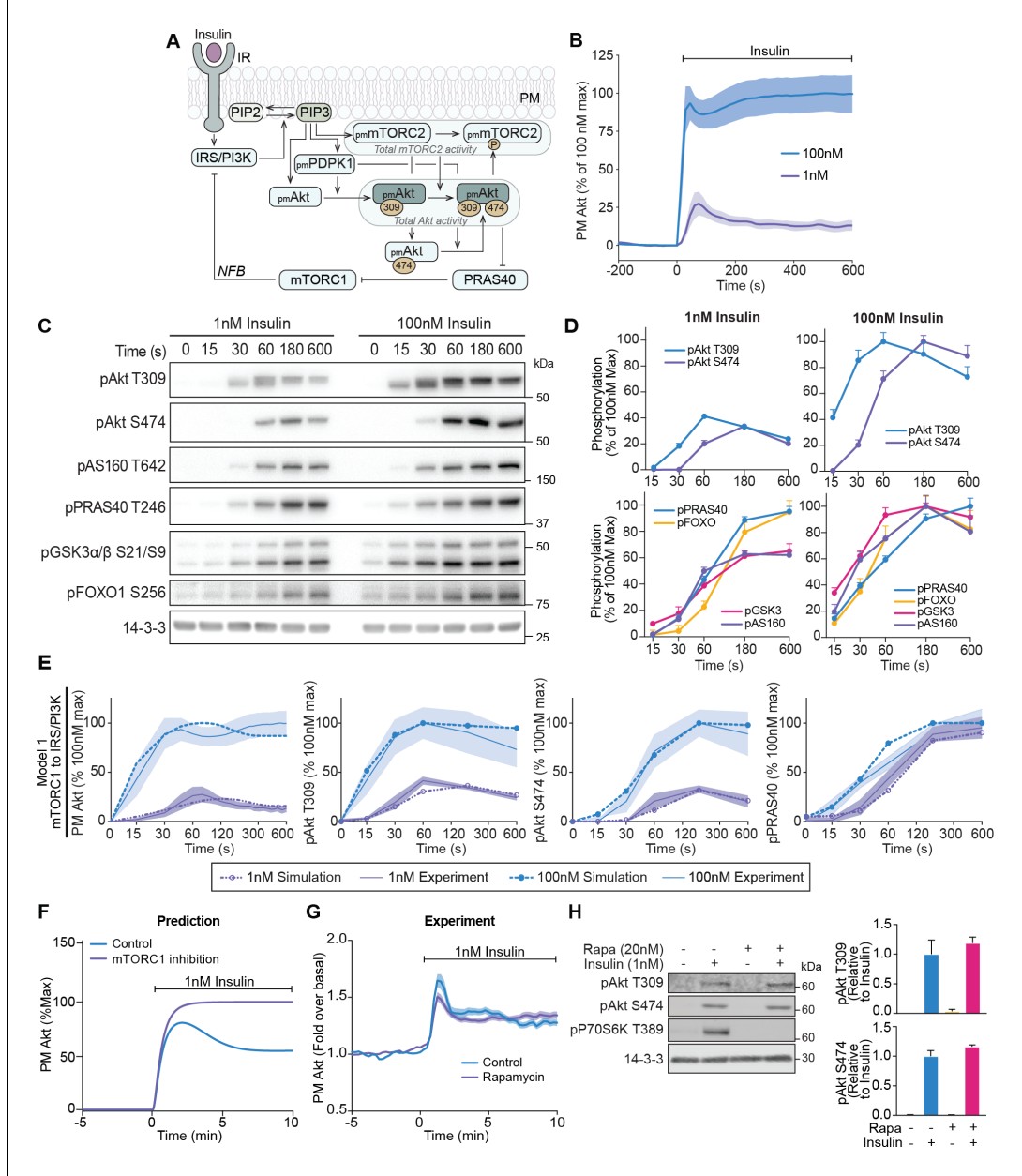

**Figure 1.** A mechanistic model of insulin signalling predicts a rapid negative feedback mechanism that is not dependent on mammalian target of rapamycin complex 1 (mTORC1). (**A**) Schematic depicting model architecture with negative feedback via mTORC1 onto the insulin receptor substrate/phosphoinositide 3-kinase (IRS/PI3K) node (IR, insulin receptor; NFB, negative feedback). (**B**) 3T3-L1 adipocytes expressing TagRFP-T-Akt2 were treated with 1 or 100 nM insulin. Recruitment was assessed using live cell total internal reflection fluorescence microscopy (TIRFM) ($n$ = 9 for 1 nM and $n$ = 10 for 100 nM). (**C**) 3T3-L1 adipocytes were treated with 1 or 100 nM insulin for the times specified. Lysates were immunoblotted with antibodies as specified, with 14-3-3 as a loading control. A representative image for five to eight independent biological replicates is presented. (**D**) Quantification of panel C. Data is normalised to 14-3-3 and expressed as a percentage of the maximum phosphorylation observed for each phospho-site in response to 100 nM insulin (1 nM; $n$ = 8, 100 nM; $n$ = 5; p, phosphorylated). (**E**) Model 1 (mTORC1 to IRS/PI3K) was calibrated (fitted) using Akt recruitment, T309 and S474 phosphorylation, and PRAS40 phosphorylation in response to 1 and 100 nM insulin (purple and blue, respectively) and overlaid with the experimentally observed kinetics for each outcome. The time axes are presented as a log$_2$ scale to highlight the early response kinetics, with 0 added for completeness. Model simulations are shown as dotted lines while experimental kinetics are shown as smooth lines as the mean ± SEM. (**F**) Model prediction of the effect of mTORC1 inhibition on Akt recruitment in response to 1 nM insulin. The value of parameter Ki2 was set to null to model complete inhibition of mTORC1 catalytic activity. (**G**) 3T3-L1 adipocytes expressing TagRFP-T-Akt2 were treated with 20 nM rapamycin for 5 min followed by 1 nM insulin. Recruitment was assessed using TIRFM (38 control cells [$n$ = 2], 39 rapamycin treated cells [$n$ = 3]). (**H**) 3T3-L1 adipocytes were treated with 20 nM rapamycin for 5 min followed by 1 nM insulin for 10 min. Lysates were immunoblotted with antibodies as specified, with 14-3-3 as a

*Figure 1 continued on next page*

*Figure 1 continued*

loading control. A representative western blot (left) and quantification (right) is presented (n = 3; p, phosphorylated). All data expressed as mean ± SEM; PM, plasma membrane; IR, insulin receptor; Rapa, rapamycin.

The online version of this article includes the following figure supplement(s) for figure 1:

**Figure supplement 1.** A mechanistic model of insulin signalling without negative feedback does not capture signalling kinetics.

## Negative feedback is not mediated via mTORC1

Our model, when trained with the Akt recruitment, Akt phosphorylation, and PRAS40 phosphorylation data, was able to reasonably recapitulate the experimental data (model 1; *Figure 1E*). We next used our model to predict the effect of a 1 nM insulin stimulus in the presence of the mTORC1 inhibitor rapamycin, to block mTORC1-mediated negative feedback. The model predicted a loss of the overshoot and a 25% increase in the recruitment of Akt to the PM (*Figure 1F*). To test this prediction, we recapitulated these conditions in 3T3-L1 adipocytes. Despite complete inhibition of mTORC1 activity by rapamycin, no change was observed in either Akt recruitment (*Figure 1G*) or phosphorylation (*Figure 1H*). When we included this rapamycin data for model calibration, the model could not reasonably recapitulate the experimental data (model 2; *Figure 1—figure supplement 1A,B*), further suggesting that mTORC1 is unlikely to be involved in negative feedback under these conditions. Furthermore, calibration of a model without negative feedback was unable to recapitulate the overshoot in Akt recruitment or phosphorylation (model 3; *Figure 1—figure supplement 1C*). Together, these data point to a feedback mechanism during the early insulin response that drives the overshoot in Akt recruitment that is independent of mTORC1.

## Negative feedback is Akt-dependent

Based on the existing model architecture, we hypothesised that negative feedback from PDPK1, mTORC2, or Akt could give rise to an overshoot in Akt recruitment when connected to an upstream node such as IRS/PI3K or PTEN. To explore these mechanisms, we constructed six additional models that each incorporated a potential negative feedback mechanism and calibrated each with our training data (models 4–9; *Figure 2A*, *Figure 2—figure supplement 1A–F*, *Supplementary file 1*). This allowed us to examine possible competing feedback structure hypotheses, by performing separate model calibrations and comparing how well each of the model variants fitted the experimental data. Identifiability analysis (*Maiwald et al., 2016*; *Rateitschak et al., 2012*; *Raue et al., 2009*) demonstrated that all models were similarly unidentifiable, with at least one parameter not identifiable in each model (*Supplementary file 2*), which is not surprising given the size and scope of the models (see Materials and methods for detailed description).

Quantitative assessment of model fitting based on the objective function revealed that Akt to IRS/PI3K (model 9) best matched the experimental data, followed by mTORC2 to PTEN (model 7), and then Akt to PTEN (model 8) (*Figure 2B*). Consistent with this, qualitative assessment of models 7–9 revealed greater consistency between model dynamics and experimental data compared to models 4–6 (*Figure 2—figure supplement 1A–F*). For example, model 4 showed a significant discordance in pAkt S474 dynamics, while models 5 and 6 displayed delayed pAkt T309 accumulation in response to 1 nM insulin (*Figure 2—figure supplement 1A–C*). Overall, since models 7–9 displayed superior quantitative and qualitative consistency with experimental data, we focused on interrogating these models hereafter.

Each of the feedback loops in these three models would be driven by Akt (Akt to IRS/PI3K, Akt to PTEN) or partially driven by Akt (mTORC2 to PTEN; as Akt contributes to mTORC2 activation [*Humphrey et al., 2013*; *Yang et al., 2015*]). Thus, we simulated the effect of Akt inhibition on insulin-stimulated Akt recruitment in the three models. In all cases it was predicted that Akt inhibition would eliminate the overshoot in Akt recruitment and facilitate increased recruitment, but with distinct kinetics and magnitude (*Figure 2C*). To test these predictions experimentally, we used TIRFM to measure the recruitment of TagRFP-T-Akt2 in the presence of the Akt inhibitor GDC0068, which inhibits Akt by binding its ATP-binding pocket (*Figure 2—figure supplement 2A*). Strikingly, in the presence of GDC0068, we observed a threefold increase in Akt recruitment to the PM upon insulin stimulation and loss of overshoot (*Figure 2D*). The recruitment kinetics and magnitude were markedly similar to the model predictions with the removal of feedback from Akt to IRS/PI3K

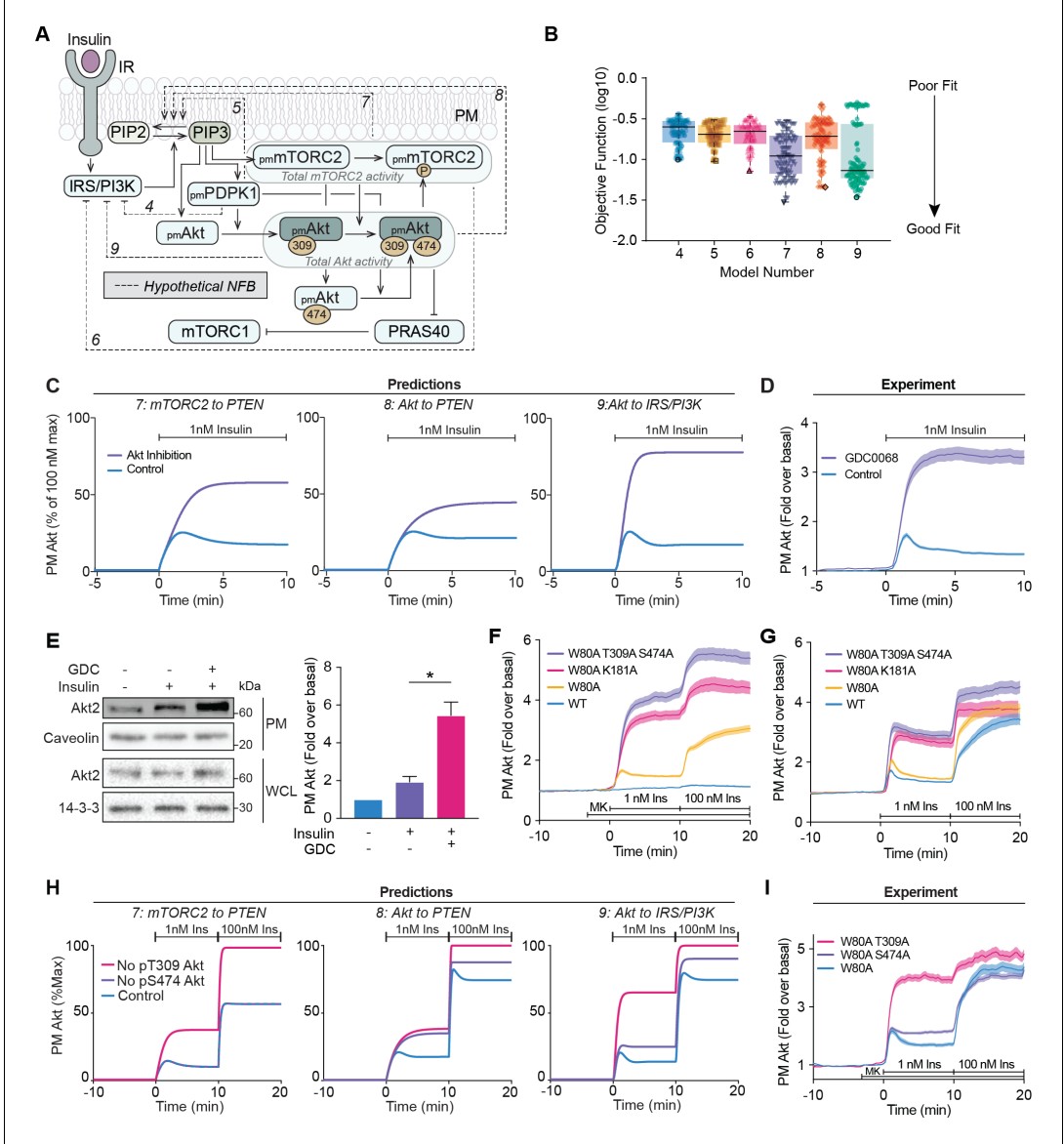

**Figure 2.** Negative feedback is dependent on Akt activity. (**A**) Schematic depicting the model architecture including the hypothetical negative feedback loops from phosphoinositide-dependent protein kinase 1 (PDPK1), mammalian target of rapamycin complex 2 (mTORC2), or Akt, onto phosphatase and tensin homolog (PTEN) or the insulin receptor substrate/phosphoinositide 3-kinase (IRS/PI3K) node (IR, insulin receptor; NFB, negative feedback). (**B**) Box and whiskers plots (median; interquartile range and min/max) of the local minima from 200 independent calibration runs (repeats) of negative feedback models 4–9, where for each run, we randomly resampled the starting parameter values as input for the genetic algorithm. The lowest of these were deemed the global optimum (dark border) and these were used for simulations and predictions. (**C**) Predictions from models 7–9 for the effect of Akt inhibition on Akt recruitment in response to 1 nM insulin. The parameter values of Ki2a, Ki2b, Kf11a, and Kf11b were set to null in order to model Akt inhibition by GDC0068. (**D**) 3T3-L1 adipocytes expressing TagRFP-T-Akt2 were treated with 10 µM GDC0068 for 5 min followed by 1 nM insulin. Recruitment was assessed using total internal reflection fluorescence microscopy (TIRFM) (41 control cells [$n$ = 3], 43 GDC0068-treated cells [$n$ = 3]). (**E**) 3T3-L1 adipocytes were treated with 10 µM GDC0068 or vehicle control for 5 min followed by 1 nM insulin for 10 min. Subcellular fractionation was performed to obtain the plasma membrane. Lysates were immunoblotted with antibodies as specified, with caveolin and 14-3-3 as loading controls. A representative western blot (left) and quantitation of plasma membrane Akt2 (right) are presented ($n$ = 6, unpaired two-tailed t-test; WCL, whole cell lysate). (**F**) 3T3-L1 adipocytes expressing WT, W80A, W80A-T309A-S474A, or W80A-K181A TagRFP-T-Akt2 were exposed to 20 µM MK2206 for 3 min, followed by 1 and 100 nM insulin. Recruitment was assessed using TIRFM (16 WT cells [$n$ = 2], 93 W80A cells [$n$ = 10], 53 W80A-K181A cells [$n$ = 4], 35 W80A-T309A-S474A cells [$n$ = 3]). (**G**) 3T3-L1 adipocytes expressing WT, W80A, W80A-T309A-S474A, or W80A-K181A TagRFP-T-Akt2 were treated with 1 and 100 nM insulin in the absence of MK2206. Recruitment was assessed using TIRFM (41 WT cells [$n$ = 3], 60 W80A cells [$n$ = 7], 21 W80A-K181A cells [$n$ = 2], 22 W80A-T309A-S474A cells [$n$ = 2]). (**H**) Model predictions of the effect of losing Akt T309 or Akt S474 phosphorylation on Akt recruitment in response to 1 and 100 nM insulin. The parameter values of Kf6 and Kf7 were set to null to represent loss of Akt T309 and Akt S474 phosphorylation,

*Figure 2 continued on next page*

*Figure 2 continued*

respectively. (I) 3T3-L1 adipocytes expressing W80A, W80A-T309A, or W80A-S474A TagRFP-T-Akt2 were exposed to 20 μM MK2206 for 3 min, followed by 1 and 100 nM insulin. Recruitment was assessed using TIRFM (55 W80A cells [*n* = 9], 57 W80A T309A cells [*n* = 6] 121 W80A S474A cells [*n* = 12]). All data expressed as mean ± SEM; PM, plasma membrane; *p<0.05.

The online version of this article includes the following figure supplement(s) for figure 2:

**Figure supplement 1.** Six hypothetical mechanistic models of insulin signalling predict signalling outcomes.

**Figure supplement 2.** Akt inhibition increases its phosphorylation and membrane recruitment following insulin stimulation, in diverse cell types.

(*Figure 2C*). These data were corroborated by assessment of endogenous Akt localisation in adipocytes using subcellular fractionation, which showed a marked potentiation of insulin-stimulated Akt2 recruitment to the PM in the presence of GDC0068 (*Figure 2E*). This phenomenon was not specific to adipocytes, but rather a common feature of multiple cell types – HEK293E, HeLa, 3T3-L1 fibroblasts, and MCF7 cells (*Figure 2—figure supplement 2B–E*).

To ensure these effects were not GDC0068-specific, but rather a general feature of Akt inhibition, we employed an orthogonal chemical genetics approach to inhibit Akt. The W80A mutation in Akt renders it insensitive to inhibition by MK2206, a compound that prevents Akt PM recruitment (*Wu et al., 2010*). Thus, MK2206 can block endogenous Akt activation, leaving ectopic W80A Akt kinase-dead mutants to be specifically interrogated in cells (*Green et al., 2008*; *Kajno et al., 2015*; *Kearney et al., 2019*). In the presence of MK2206, W80A TagRFP-T-Akt2 displayed similar recruitment kinetics to WT TagRFP-T-Akt2 (*Figure 2F*), consistent with our earlier study (*Kearney et al., 2019*). We assessed insulin-stimulated recruitment of two W80A kinase-dead TagRFP-T-Akt2 mutants, which confer loss of kinase activity by differential mechanisms; by preventing Akt phosphorylation at its activating sites (W80A-T309A-S474A; *Beg et al., 2017*), or by preventing ATP binding (W80A-K181A; *Cong et al., 1997*). Both mutants exhibited 3.5-fold augmented recruitment and loss of overshoot behaviour compared to W80A Akt (*Figure 2F*). These responses phenocopied the recruitment of WT Akt in the presence of GDC0068 (*Figure 2—figure supplement 2F*), and again were remarkably similar to the model prediction with the removal of feedback from Akt to IRS/PI3K (*Figure 2C*). In the absence of MK2206, the PM recruitment of the kinase-dead mutants was still potentiated, but also exhibited an overshoot, suggesting competition with endogenous Akt (*Figure 2G*). Taken together, these data indicate the presence of an Akt-dependent feedback signal which limits its activation at the PM, likely through the IRS/PI3K node.

## Phosphorylation of Akt at S474 is not required for negative feedback engagement

While phosphorylation of Akt at T309 is essential for Akt kinase activity (*Alessi et al., 1996a*; *Kearney et al., 2019*), the role of S474 phosphorylation remains controversial. However, in adipocytes, we have shown S474 phosphorylation is required only for maximal kinase activity (*Kearney et al., 2019*). As the overshoot in Akt recruitment correlated with the kinetics of T309 phosphorylation, but not S474 phosphorylation (*Figure 2—figure supplement 2G*), we hypothesised that feedback is rapidly engaged following Akt T309 phosphorylation, and that S474 phosphorylation is not required for feedback engagement.

To test this, we simulated the effect of losing either T309 or S474 phosphorylation on insulin-stimulated Akt recruitment in our three models (*Figure 2H*). As expected, the predictions for loss of T309 phosphorylation in each model was identical to their prediction of Akt inhibition (*Figure 2C*). However, each model's prediction for loss of S474 phosphorylation was markedly different (*Figure 2H*). We tested these predictions experimentally by observing insulin-stimulated recruitment of W80A-T309A and W80A-S474A TagRFP-T-Akt2 by TIRFM (*Figure 2I*). Loss of S474 phosphorylation resulted in an attenuated overshoot in response to 1 nM insulin but no difference in response magnitude compared to control. This recruitment profile is consistent with a subtle loss of feedback, which likely occurs because Akt is only fully active once phosphorylated at both T309 and S474 residues (*Alessi et al., 1996a*; *Kearney et al., 2019*). An additional twofold increase was observed with 100 nM insulin, similar to control. In response to 1 nM insulin, W80A-T309A Akt displayed a loss of overshoot and a threefold increase in PM association in comparison to control (*Figure 2I*). This phenocopied chemical and genetic inhibition of Akt (*Figure 2D,F*), which was

expected since T309A Akt has no kinase activity (*Alessi et al., 1996a*; *Kearney et al., 2019*). Additionally, with a subsequent 100 nM insulin stimulus, there was only a modest increase in W80A-T309A recruitment. These response profiles were in close agreement with the predictions arising from the Akt to IRS/PI3K model (*Figure 2H*). These combined modelling and experimental analyses support that the phosphorylation of Akt at S474 is largely redundant in activating the negative feedback.

## Akt regulates PI(3,4,5)P3 abundance

Irrespective of where the feedback was engaged, all three models assumed that Akt regulates PIP3 levels through negative feedback. To assess whether this was the case, we measured insulin-stimulated PIP3 content in adipocytes, in the presence or absence of Akt inhibition. We utilised three small-molecule inhibitors, which abolish Akt kinase activity via distinct mechanisms: by binding the ATP-binding site of Akt (GDC0068), by preventing Akt PM recruitment (MK2206), and by inhibiting PDPK1 to prevent Akt T309 phosphorylation (GSK2334470) (*Figure 3—figure supplement 1*). Stimulation with 1 nM insulin increased PIP3 content approximately fivefold relative to basal, confirming the validity of the assay (*Figure 3A*). Strikingly, each inhibitor further augmented insulin-stimulated PIP3 abundance approximately fivefold relative to insulin alone (*Figure 3A*), suggesting that Akt regulates PIP3 abundance.

We hypothesised that the ability of Akt to regulate PIP3 abundance would influence the recruitment of other PIP3-binding proteins, such as PDPK1 and GAB2 (*Currie et al., 1999*; *Gu et al., 2003*; *Yoshizaki et al., 2007*). Both PDPK1-eGFP and Gab2PH-eGFP were recruited to the PM upon insulin stimulation and displayed PM recruitment overshoots (*Figure 3B,C*). Furthermore, their degree of recruitment was markedly enhanced with GDC0068 and MK2206 (*Figure 3B–D*), consistent with increased PIP3 in the absence of Akt activity. To directly assess the requirement of Akt activity for feedback, we examined recruitment of Gab2PH-eGFP in cells co-expressing WT, W80A, or W80A-T309A TagRFP-T-Akt2. In the presence of MK2206, the Akt constructs behaved as expected (*Figure 3E*), while MK2206-dependent hyper-recruitment of Gab2PH-eGFP was rescued by co-expression of W80A Akt, but not WT or W80A-T309A Akt (*Figure 3F*). These data suggest that Akt regulates PIP3 abundance and this alters the localisation of PIP3-binding proteins.

## Akt regulates PI3K recruitment to the PM

Elevated PIP3 abundance following Akt inhibition likely resulted from its increased production by PI3K, or suppressed breakdown by phosphatases such as PTEN. So far, the Akt to IRS/PI3K model best recapitulated the experimental data; however to further interrogate this model, we tested whether Akt controls PIP3 degradation. To this end, we simulated the effect of PI3K inhibition on the rate of Akt dissociation from the PM across the three models. Both the Akt to PTEN and mTORC2 to PTEN models predicted a slower rate of Akt dissociation in the absence of Akt activity, due to impaired PIP3 degradation by PTEN (*Figure 4A*). In contrast, the Akt to IRS/PI3K model predicted no difference in Akt PM dissociation (*Figure 4A*). To test this experimentally, we stimulated adipocytes expressing W80A (active) or W80A-T309A (kinase-dead) TagRFP-T-Akt2 with insulin, and then the PI3K inhibitor wortmannin, once PM Akt had achieved a steady-state concentration. There was no difference in their rate of PM dissociation, consistent with the Akt to IRS/PI3K negative feedback model (*Figure 4B*). We next conducted the same experiment using PDPK1-eGFP, in the presence or absence of GDC0068 or MK2206. Regardless of whether Akt was inhibited, there was no difference in the rate of PDPK1 PM dissociation (*Figure 4C*). To corroborate these experiments, we used adipocytes expressing constitutively active PI3K (p110*), as well as W80A (active) or W80A-T309A (kinase-dead) TagRFP-T-Akt2. Expression of p110* (*Hu et al., 1995*) allowed a constant rate of PIP3 production, while the rate of PIP3 degradation was unaltered. When stimulated with MK2206, there was no increase in W80A-T309A Akt recruitment, indicating no loss of feedback (*Figure 4D*). Furthermore, there was no difference in the PM dissociation rate between W80A and W80A-T309A Akt (*Figure 4D*). We next measured TagRFP-T-Akt2 recruitment in HCC1937 human breast cancer cells, which do not express PTEN, the primary PIP3 phosphatase (*Kabuyama et al., 1996*; *Lee et al., 2018*; *Maehama and Dixon, 1998*; *Myers et al., 1998*). Akt recruitment increased in response to IGF1 and increased further with GDC0068 (*Figure 4E*). This was consistent with the engagement of Akt-mediated feedback in the absence of PTEN.

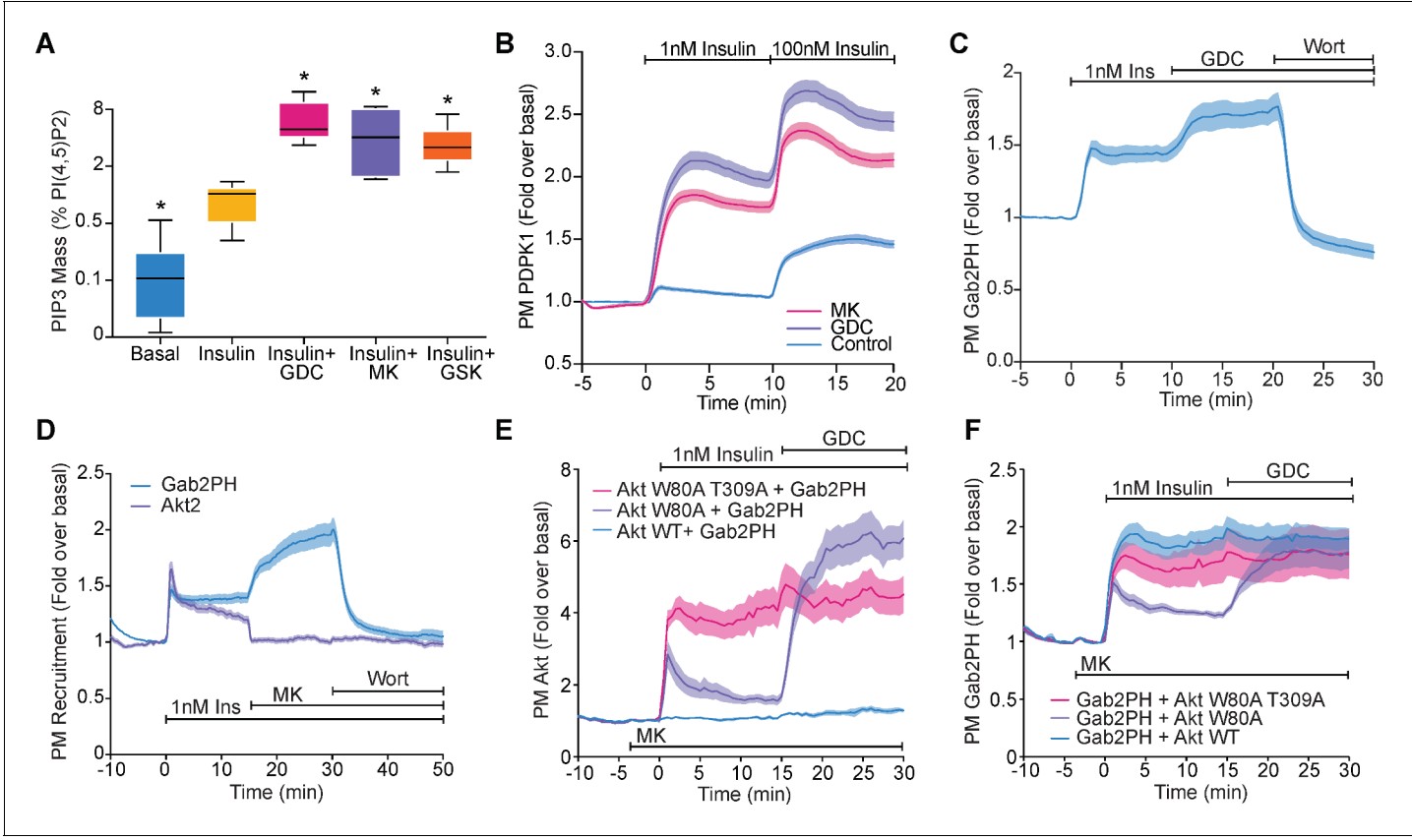

**Figure 3.** Akt regulates PI(3,4,5)P3 abundance. (**A**) 3T3-L1 adipocytes were treated with 10 μM GDC0068, 10 μM MK2206, 10 μM GSK2334470, or vehicle control for 15 min, followed by 1 nM insulin for 10 min. Lipids were extracted from the cells and PI(3,4,5)P3 (PIP3) mass measured using a competitive ELISA. PIP3 is expressed relative to PI(4,5)P2 obtained from the same samples ($n$ = 6–7). Boxes capture lower quartile and upper quartile with the median displayed as a horizontal line in the middle; whiskers are min and max. (**B**) 3T3-L1 adipocytes expressing phosphoinositide-dependent protein kinase 1 (PDPK1)-eGFP were stimulated with 1 and 100 nM insulin, in the presence of 10 μM GDC0068, 10 μM MK2206, or vehicle control. Recruitment was assessed by total internal reflection fluorescence microscopy (TIRFM) (123 control cells [$n$ = 3], 121 GDC0068-treated cells [$n$ = 3], 118 MK2206-treated cells [$n$ = 3]). (**C**) 3T3-L1 adipocytes expressing Gab2PH-eGFP were stimulated with 1 nM insulin, followed by 10 μM GDC0068, and then 1 μM wortmannin. Recruitment was assessed by TIRFM (11 cells, $n$ = 3). (**D**) 3T3-L1 adipocytes co-expressing TagRFP-T-Akt2 and Gab2PH-eGFP were stimulated with 1 nM insulin, 10 μM MK2206 and then 1 μM wortmannin. Recruitment was assessed by TIRFM (20 cells, $n$ = 4). (**E**) TagRFP-T-Akt2 recruitment responses and (**F**) the corresponding Gab2PH-eGFP recruitment responses of 3T3-L1 adipocytes co-expressing Gab2PH-eGFP with either WT, W80A, or W80A-T309A TagRFP-T-Akt2. Cells were treated with 10 μM MK2206 3 min prior to 1 nM insulin and then 10 μM GDC0068 (7, 8, and 9 cells expressing WT, W80A, and W80A T309A Akt2-TagRFP-T, respectively; $n$ = 3). All data expressed as mean ± SEM; PM, plasma membrane; *$p<0.05$ compared to insulin alone.

The online version of this article includes the following figure supplement(s) for figure 3:

**Figure supplement 1.** GSK2334470 inhibits phosphoinositide-dependent protein kinase 1 (PDPK1)-dependent signalling outcomes in 3T3-L1 adipocytes.

Both our modelling and experimental data suggested that Akt regulates the ability of PI3K to synthesise PIP3. As PI3K is activated at the PM, we determined whether Akt controls PI3K localisation. Subcellular fractionation revealed an increase in the abundance of PI3K p85 and p110 at the PM upon insulin stimulation, which was potentiated in the presence of Akt inhibitors GDC0068 and MK2206 (*Figure 4F*). These data were corroborated by assessment of PI3K p85 localisation using immunofluorescence/TIRFM. This technique was less sensitive than subcellular fractionation, as it did not reveal a change in PM localised PI3K with 1 nM insulin; however, an increase was detectable in the presence of the Akt inhibitors GDC0068 and MK2206 (*Figure 4G*).

Together, these data are consistent with the Akt to IRS/PI3K negative feedback model. Specifically, upon acute growth factor stimulation, Akt translocates to the PM where it is phosphorylated

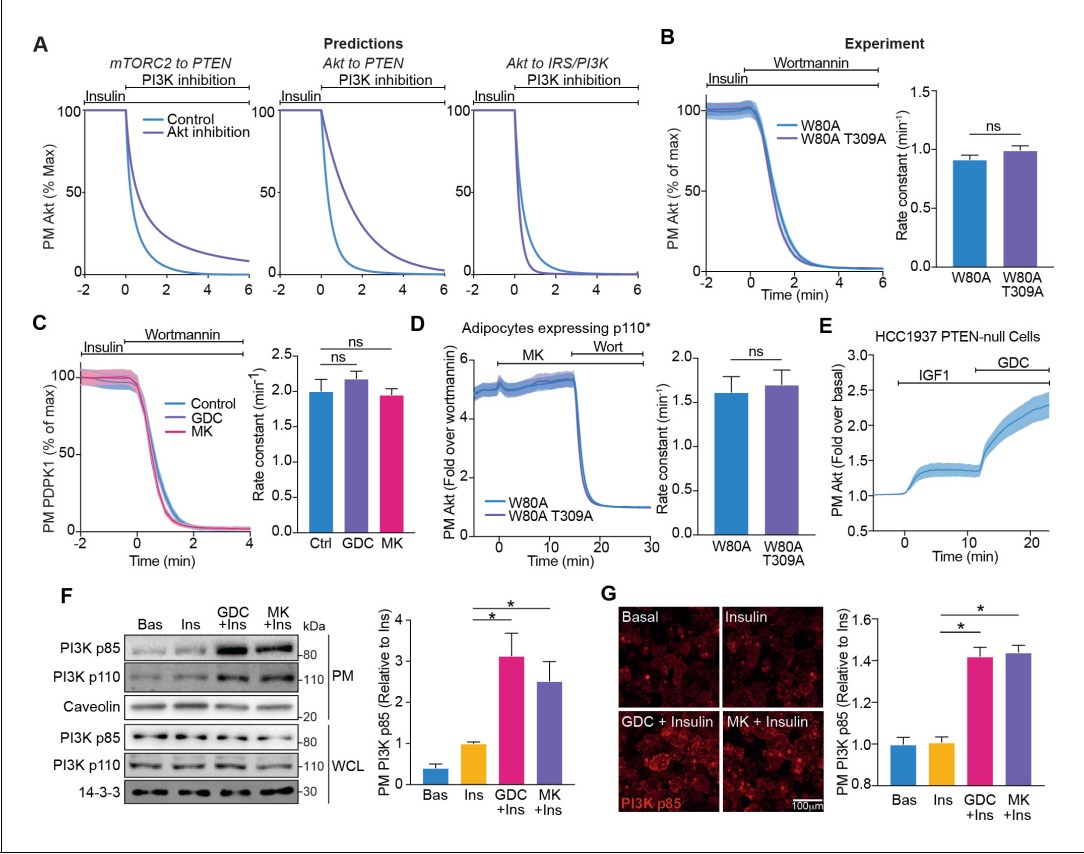

**Figure 4.** Akt regulates insulin-stimulated production of PI(3,4,5)P3 by phosphoinositide 3-kinase (PI3K). (**A**) Model predictions detailing the effect of PI3K inhibition following insulin stimulation, on the plasma membrane dissociation of Akt with and without Akt inhibition. The parameter value of Kf3 was set to null to simulate PI3K inhibition by wortmannin. (**B**) In the presence of 10 μM MK2206, 3T3-L1 adipocytes expressing W80A or W80A-T309A TagRFP-T-Akt2 were treated with 100 nM insulin, followed by 1 μM wortmannin. Plasma membrane dissociation upon wortmannin treatment was assessed using total internal reflection fluorescence microscopy (TIRFM) (left). The rate constants per individual cell (right) were calculated (70 W80A cells [*n* = 3], 65 W80A-T309A cells [*n* = 3], unpaired two-tailed t-test). (**C**) In the presence of 10 μM MK2206, 10 μM GDC0068 or vehicle control, 3T3-L1 adipocytes expressing phosphoinositide-dependent protein kinase 1 (PDPK1)-eGFP were treated with 100 nM insulin, followed by 1 μM wortmannin. Plasma membrane dissociation upon wortmannin stimulation was assessed using TIRFM (left). The rate constants per individual cell (right) were calculated (123 control cells [*n* = 3], 121 GDC0068-treated cells [*n* = 3], 118 MK2206-treated cells [*n* = 3], unpaired two-tailed t-test corrected for multiple comparisons). (**D**) 3T3-L1 adipocytes expressing p110* (constitutively active PI3K) and W80A or W80A-T309A TagRFP-T-Akt2 were treated with 10 μM MK2206 followed by 1 μM wortmannin. Plasma membrane dissociation upon wortmannin stimulation was assessed using TIRFM (left). The rate constants per individual cell (right) were calculated (108 W80A cells [*n* = 3], 98 W80A-T309A cells [*n* = 3], unpaired two-tailed t-test). (**E**) HCC1937 phosphatase and tensin homolog (PTEN)-null, human breast cancer cells expressing TagRFP-T-Akt2 were treated with 50 ng/mL IGF1, followed by 10 μM GDC0068. Recruitment was assessed using TIRFM (64 cells [*n* = 2]). (**F**) 3T3-L1 adipocytes were treated with 10 μM MK2206, 10 μM GDC0068, or vehicle control for 5 min, followed by 1 nM insulin for 10 min. The plasma membrane was obtained by subcellular fractionation and immunoblotted with antibodies as specified, with caveolin and 14-3-3 as loading controls. A representative blot is presented (left). Quantification of PM PI3K p85 is presented (right; *n* = 6, unpaired two-tailed t-test corrected for multiple comparisons; WCL, whole cell lysate). (**G**) 3T3-L1 adipocytes were treated with 10 μM MK2206, 10 μM GDC0068, or a vehicle control for 5 min, followed by 1 nM insulin for 10 min. Cells were fixed, stained with an antibody against PI3K p85 (immunofluorescence), and imaged using TIRFM. A representative image is presented. Scale bar represents 100 μm. Relative p85 signal was quantified (129 basal cells [*n* = 2], 129 insulin-treated cells [*n* = 2], 129 GDC0068/insulin-treated cells [*n* = 2], 158 MK2206/insulin-treated cells [*n* = 2], unpaired two-tailed t-test corrected for multiple comparisons). All data expressed as mean ± SEM; PM, plasma membrane; ns, not significant; *p<0.05.

and activated. Akt then engages a negative feedback mechanism that limits PM-associated PI3K and consequently lowers PIP3 abundance.

## Akt releases IRS from the PM to the cytosol to limit IR-IRS interaction

We next investigated how Akt controls PI3K localisation. The speed of the feedback suggested that Akt-mediated phosphorylation of PI3K itself could drive changes in its localisation. However, PI3K does not contain an Akt substrate motif (R-X-R-X-X-S/T, where R represents arginine, X represents

any amino acid, and S/T represents serine/threonine) (*Alessi et al., 1996b*). Furthermore, translocation of PI3K to the PM is thought to be primarily regulated by interactions with other proteins, rather than post-translational modification of PI3K itself (*Rordorf-Nikolic et al., 1995*). IRS1 and IRS2 are the primary adaptor proteins responsible for recruiting PI3K to the PM following insulin stimulation (*White, 2003*). Consequently, we explored whether Akt regulates IRS1/2 behaviour.

First, we determined whether Akt controls the abundance of IRS1-eGFP and IRS2-eGFP at the PM using TIRFM. Following insulin stimulation, the amount of IRS1/2 at the PM decreased (*Figure 5A,B*). However, in the presence of Akt inhibitors (GDC0068 and MK2206), IRS1/2 were retained at the PM (*Figure 5A,B*). Importantly, rapamycin was unable to inhibit the insulin-stimulated decrease in PM-associated IRS1/2 (*Figure 5C,D*), demonstrating that this process is mTORC1/S6K-independent. Consistent with these observations, subcellular fractionation also demonstrated an Akt-dependent decrease in the abundance of endogenous IRS1 at the PM and extended these findings to reveal that IRS1 moves from the PM to the cytosol upon insulin stimulation, rather than to internal membranes (high density microsome [HDM]/low density microsome [LDM] fractions) (*Figure 5E*). We attribute the PM localisation of IRS1/2 prior to insulin stimulation to its PH domain, as deletion of the IRS1 PH domain (DelPH IRS1-eGFP) decreased its abundance at the PM in unstimulated cells (*Figure 5F*). This is likely due to the affinity of the IRS1 PH domain for PI(4,5)P2 (*Dhe-Paganon et al., 1999*). Together, these data indicate that IRS1/2 is PM-localised in unstimulated cells via a PH domain-dependent interaction, and upon insulin stimulation Akt induces the translocation of IRS1/2 from the PM to the cytosol.

We next dissected whether these changes in IRS1/2 localisation were a product of Akt modulating the localisation or activation of the IR. Consistent with direct regulation of IRS by Akt, treatment with Akt inhibitors did not alter IR localisation (*Figure 5G*) or activation as measured by tyrosine phosphorylation (*Figure 5H*). Furthermore, deletion of the PTB domain of IRS1 (DelPTB IRS1-eGFP), which is responsible for its interaction with the IR (*Eck et al., 1996*), did not impact IRS1 removal from the PM upon insulin stimulation (*Figure 5I*). These data suggest that Akt removes IRS from the PM via disruption of IR-independent interactions.

Despite occurring independent of the IR, we hypothesised that these changes in IRS localisation would ultimately serve to shrink the pool of IRS available to interact with activated IR. Following immunoprecipitation of endogenous IRS1 or IRS2 followed by mass spectrometry, the IR was only detected in the presence of insulin, consistent with a strong insulin-dependent interaction (*Figure 5J*). However, the amount of IR detected was augmented in the presence of Akt inhibitors GDC0068 and MK2206 (approximately 17- and 7-fold for IRS1; 19- and 12-fold for IRS2, respectively; *Figure 5J*), suggesting that Akt controls the level of interaction between the IR and IRS. These data suggest that Akt limits PI3K-mediated PIP3 production by depleting PM-associated IRS, to reduce the pool of IRS available to interact with activated IR.

## Phosphorylation of IRS releases it from the PM and limits Akt recruitment

We next explored how Akt controls IRS localisation. Phosphorylation is frequently responsible for changes in protein localisation (*Cohen, 2002*) and we have previously identified an abundance of insulin-regulated phosphorylation sites on IRS1/2 (*Humphrey et al., 2013*). Thus, we hypothesised that Akt phosphorylates IRS to release it from the PM and engage negative feedback.

Collectively, IRS1 and IRS2 contained 11 phosphorylation sites within an Akt substrate motif (R-X-R-X-X-S/T) (*Alessi et al., 1996b*). To investigate whether these phosphorylation sites were responsible for the depletion of PM-associated IRS1/2 upon insulin stimulation, we concurrently mutated these Ser/Thr residues on human IRS1-eGFP (herein 6P IRS1) and human IRS2-eGFP (herein 5P IRS2) to alanine, to prevent phosphorylation (*Figure 6A,B*). We then co-expressed 6P IRS1 or 5P IRS2 with TagRFP-T-Akt2 in adipocytes and utilised TIRFM to measure changes in their localisation. Following insulin stimulation, WT IRS1/2 dissociated from the PM; however, increased PM abundance was observed for both 6P IRS1 and 5P IRS2 (*Figure 6C,D*). Furthermore, expression of 6P IRS1 or 5P IRS2 in adipocytes resulted in hyper-recruitment of Akt to the PM compared to cells expressing WT IRS (*Figure 6C,D*). This response mimicked GDC0068 treatment (*Figure 2D*), causing athreefold increase in PM Akt upon insulin stimulation. These data suggest that phosphorylation of IRS1/2 facilitates its removal from the PM and drives negative feedback.

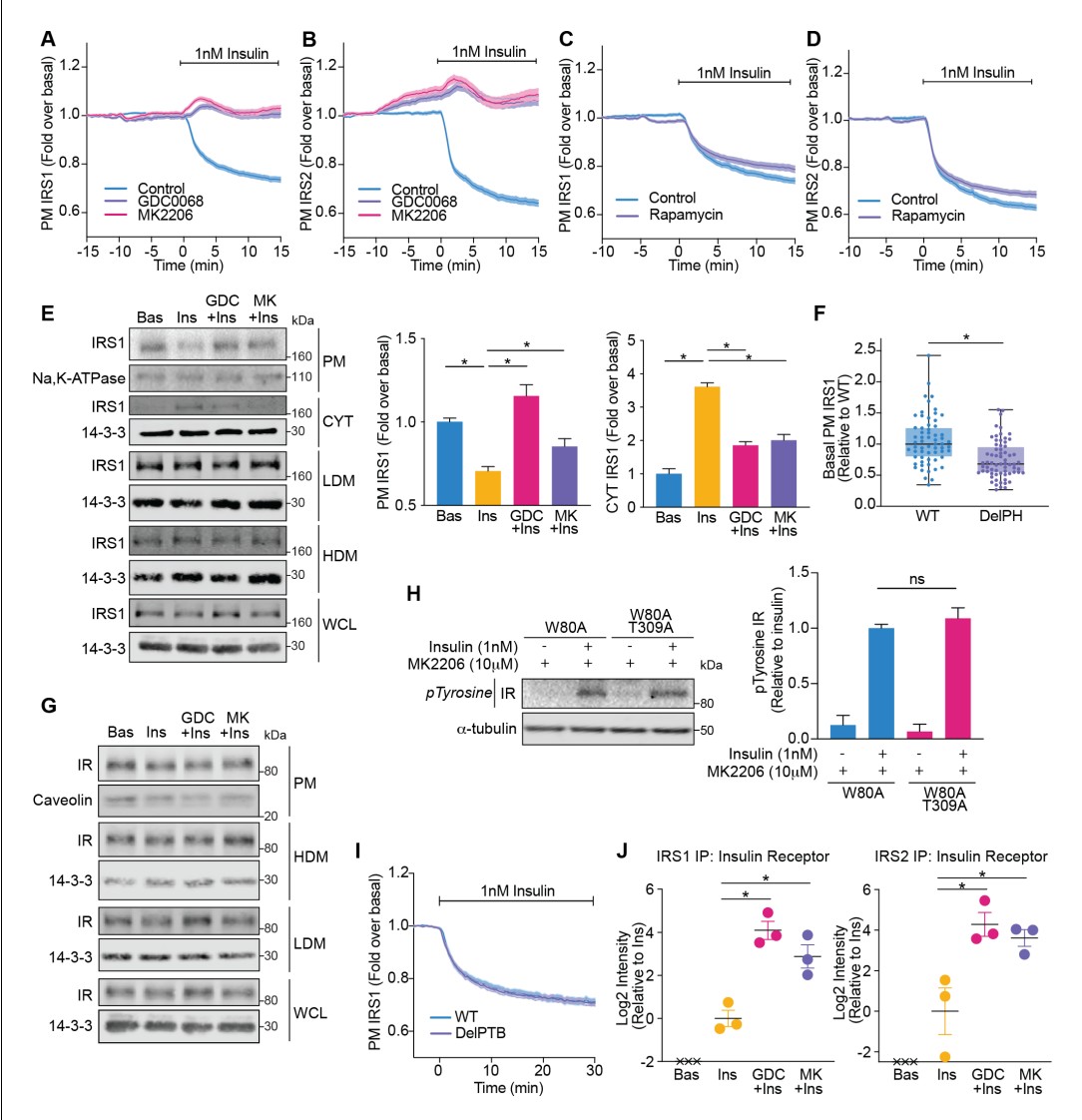

**Figure 5.** Akt releases insulin receptor substrate 1/2 (IRS1/2) from the plasma membrane to the cytosol. (**A**) 3T3-L1 adipocytes expressing IRS1-eGFP were stimulated with 10 μM GDC0068, 10 μM MK2206, or vehicle control for 10 min, followed by 1 nM insulin. Recruitment was assessed by total internal reflection fluorescence microscopy (TIRFM) (85 control cells [$n = 3$], 68 GDC0068-treated cells [$n = 3$], 79 MK2206-treated cells [$n = 3$]). (**B**) 3T3-L1 adipocytes expressing IRS2-eGFP were stimulated with 10 μM GDC0068, 10 μM MK2206, or vehicle control for 10 min, followed by 1 nM insulin. Recruitment was assessed by TIRFM (71 control cells [$n = 3$], 73 GDC0068-treated cells [$n = 3$], 69 MK2206-treated cells [$n = 3$]). (**C**) 3T3-L1 adipocytes expressing IRS1-eGFP were stimulated with 20 nM rapamycin or vehicle control for 5 min, followed by 1 nM insulin. Recruitment was assessed by TIRFM (70 control cells [$n = 3$], 75 rapamycin treated cells [$n = 3$]). (**D**) 3T3-L1 adipocytes expressing IRS2-eGFP were stimulated with 20 nM rapamycin or vehicle control for 5 min, followed by 1 nM insulin. Recruitment was assessed by TIRFM (66 control cells [$n = 3$], 77 rapamycin treated cells [$n = 3$]). (**E**) 3T3-L1 adipocytes were treated with 10 μM GDC0068, 10 μM MK2206, or vehicle control for 5 min followed by 1 nM insulin for 10 min. Subcellular fractionation was performed and lysates were immunoblotted with antibodies as specified, with Na,K-ATPase and 14-3-3 as loading controls. A representative western blot is presented (WCL, whole cell lysate; Cyt, cytosol; LDM, low density microsomes; HDM, high density microsomes). Quantification of PM IRS1 ($n = 4$, two-tailed t-test corrected for multiple comparisons) and Cyt IRS1 ($n = 3$, two-tailed t-test corrected for multiple comparisons) is presented. (**F**) Unstimulated 3T3-L1 adipocytes expressing WT or DelPH (deletion of pleckstrin homology [PH] domain, residues 2–115 removed) IRS1-eGFP were imaged by TIRF and epifluorescence microscopy. The median TIRF intensity of each cell was normalised to its median epifluorescence intensity (62 WT cells [$n = 3$], 67 DelPH cells [$n = 3$], unpaired two-tailed t-test, each circle represents 1 cell). (**G**) 3T3-L1 adipocytes were treated with 10 μM GDC0068, 10 μM MK2206, or vehicle control for 5 min followed by 1 nM insulin for 10 min. Subcellular fractionation was performed and lysates were immunoblotted with antibodies as specified, with caveolin and 14-3-3 as loading controls. A representative western blot for two independent experiments is presented (WCL, whole cell lysate; LDM, low density microsomes; HDM, high density microsomes). (**H**) 3T3-L1 adipocytes stably expressing FLAG-W80A or FLAG-W80A-T309A Akt2 were incubated with 10 μM MK2206 for 5 min followed by 1 nM insulin for 10 min. Lysates were immunoblotted with total phosphorylated tyrosine antibody (Cell Signaling Technology CST8954) and α-tubulin as a loading control (p,

*Figure 5 continued on next page*

*Figure 5 continued*

phosphorylated; IR, insulin receptor). A representative western blot is presented above. Quantification of phosphorylated tyrosine (pY) IR is presented below (n = 3, unpaired two-tailed t-test). (**I**) 3T3-L1 adipocytes expressing WT or DelPTB (deletion of phosphotyrosine-binding (PTB) domain, residues 161–265 removed) IRS1-eGFP were stimulated with 1 nM Insulin. Recruitment was assessed by TIRFM (63 WT cells [n = 3], 68 DelPTB cells [n = 3]). (**J**) 3T3-L1 adipocytes were stimulated with 10 μM GDC0068, 10 μM MK2206, or vehicle control for 5 min, followed by 1 nM insulin for 10 min. Endogenous IRS1 or IRS2 was immunoprecipitated and insulin receptor quantified by mass spectrometry (n = 3, crosses represent missing values, two-tailed t-test corrected for multiple comparisons). All data expressed as mean ± SEM; PM, plasma membrane; ns, not significant; *p<0.05.

To investigate which of these 11 phosphorylation sites were responsible for the dissociation of IRS from the PM, we individually mutated each of these Ser/Thr residues on IRS1-eGFP and IRS2-eGFP to alanine to prevent phosphorylation. We then co-expressed this mutant with TagRFP-T-Akt2 and monitored their localisation using TIRFM. No single mutation phenocopied 6P IRS1 or 5P IRS2, with each mutant having only a subtle effect on Akt and IRS localisation (*Figure 6—figure supplement 1A–F*, *Figure 6—figure supplement 2A–E*). This indicated that at least two insulin-stimulated phosphorylation events cooperate to release IRS1/2 from the PM and limit downstream signal propagation.

## Akt directly phosphorylates IRS2 at S306 and S577

We next investigated whether any of the 11 IRS phosphorylation sites making up the 6P IRS1 and 5P IRS2 were directly phosphorylated by Akt. Our criteria for an Akt substrate were that the IRS residue (1) can be phosphorylated by Akt in vitro, (2) cannot be phosphorylated in the presence of GDC0068/MK2206 (Akt inhibitors) in cells, and (3) can be phosphorylated in the presence of rapamycin/rapalink (mTORC1 inhibitors) in cells, to exclude S6K as the upstream kinase – S6K is activated downstream of Akt and also recognises the R-X-R-X-X-S/T substrate motif (*Alessi et al., 1996b*). In this context IRS residues phosphorylated by S6K were not of interest as this feedback mechanism is mTORC1/S6K-independent (*Figures 1* and *5C–D*).

To address these criteria, we performed three experiments which relied on quantifying IRS1/2 phosphorylation using mass spectrometry. Quantifying all 11 IRS1/2 phosphorylation sites across all experiments was technically challenging, due to limited sequence coverage. However, four of the five IRS2 phosphorylation sites were identified in all experiments, and so we focused on these phosphorylation sites hereafter. IRS2 S365 (S362 in mouse) was phosphorylated by Akt in vitro and sensitive to GDC0068/MK2206 in cells (*Figure 6—figure supplement 3A*). However, IRS2 S365 phosphorylation was also rapalink/rapamycin sensitive (*Figure 6—figure supplement 3A*), implicating it as an S6K substrate. IRS2 S1149 (S1138 in mouse) phosphorylation was not increased following insulin stimulation and was unable to be phosphorylated by Akt in vitro (*Figure 6—figure supplement 3B*), suggesting it is not an Akt substrate. However, IRS2 S306 (S303 in mouse) and S577 (S573 in mouse) were phosphorylated by Akt in vitro, and in cells were sensitive to GDC0068/MK2206, but not rapamycin/rapalink (*Figure 6E,F*). These data were sufficient to classify IRS2 S306 and S577 as novel Akt substrates.

## Akt phosphorylation of IRS2 at S306 and S577 promotes PM dissociation of IRS2 to limit downstream signal propagation

We hypothesised that Akt-mediated phosphorylation of IRS2 S306 and S577 synergistically depletes PM-associated IRS2 to drive negative feedback. When IRS2 S306 or S577 were individually mutated to alanine (S306A IRS2 or S577A IRS2), IRS2 still dissociated from the PM upon insulin stimulation, but with a higher endpoint compared to WT IRS2, and both mutants subtly increased Akt translocation to the PM (*Figure 6—figure supplement 2A,D*). However, concurrent mutation of both phosphorylation sites (S306/577A IRS2) prevented PM dissociation of IRS2 (*Figure 6G*), mimicking Akt inhibition with GDC0068 and MK2206 (*Figure 5A,B*). Furthermore, expression of S306/577A IRS2 resulted in hyper-recruitment of Akt and PDPK1 to the PM compared to cells expressing WT IRS2 (*Figure 6G,H*), consistent with increased PIP3 levels. Intriguingly, mutation of the corresponding phosphorylation sites in IRS1 (S270/527A) did not prevent its dissociation from the PM upon insulin stimulation and only had a subtle effect on Akt recruitment (*Figure 6—figure supplement 4*), demonstrating alternate regulation between the IRS isoforms. These data suggest that Akt-mediated

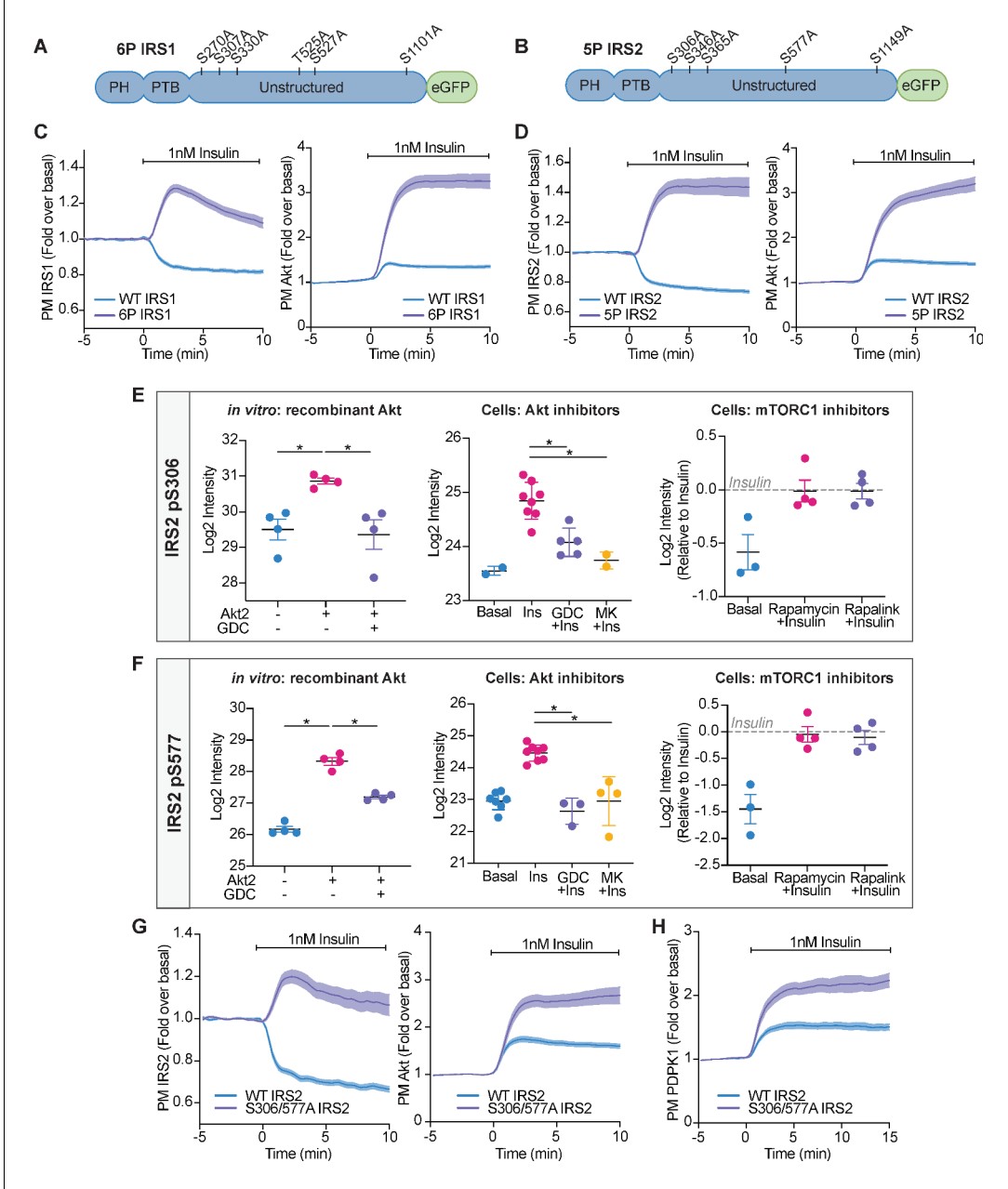

**Figure 6.** Akt-mediated phosphorylation of insulin receptor substrate (IRS) removes it from the plasma membrane to engage negative feedback. (A) IRS1-eGFP was concurrently mutated at six phosphorylation sites to make up '6P IRS1' – S270A, S307A, S330A, T525A, S527A, and S1101A (human). (B) IRS2-eGFP was concurrently mutated at five phosphorylation sites to make up '5P IRS2' – S306A, S346A, S365A, S577A, S1149A (human). Each of these Ser/Thr residues resided within an Akt substrate motif (R-X-R-X-X-S/T). Each of these residues was located outside the PH (pleckstrin homology) and PTB (phosphotyrosine-binding) domains of IRS1/2. (C) 3T3-L1 adipocytes were co-electroporated with IRS1-eGFP (WT or 6P) and TagRFP-T-Akt2. Cells were stimulated with 1 nM insulin and recruitment assessed by total internal reflection fluorescence microscopy (TIRFM) (95 WT cells [*n* = 3], 96 6 P cells [*n* = 3]). (D) 3T3-L1 adipocytes were co-electroporated with IRS2-eGFP (WT or 5P) and TagRFP-T-Akt2. Cells were stimulated with 1 nM insulin and recruitment assessed by TIRFM (139 WT cells [*n* = 5], 103 5P cells [*n* = 5]). (E) Quantification of phosphorylated IRS2 S306 (S303 in mouse) peptides across three mass spectrometry experiments. (F) Quantification of phosphorylated IRS2 S577 (S573 in mouse) peptides across three mass spectrometry experiments. (E–F) In vitro: recombinant Akt; mass spectrometry was used to quantify IRS2 phosphorylation following an in vitro assay using immunoprecipitated IRS2-FLAG and recombinant active Akt2 (*n* = 4, two-tailed t-test corrected for multiple comparisons). Cells: Akt inhibitors; 3T3-L1 adipocytes were treated with 10 µM GDC0068, 10 µM MK2206, or vehicle control for 5 min followed by 1 nM insulin for 10 min. Mass spectrometry-based phosphoproteomics was used to quantify IRS2 phosphorylation (*n* = 8, two-tailed t-test corrected for multiple comparisons). Cells: mTORC1 inhibitors; Triple-SILAC labelled HEK-293E cells were treated with 100 nM rapamycin, 3 nM rapalink, or vehicle control for 4 hr followed by 100 nM insulin for 10 min. Mass spectrometry-based phosphoproteomics was used to quantify IRS2 phosphorylation (*n* = 4). (G) 3T3-L1 adipocytes were co-

*Figure 6 continued on next page*

*Figure 6 continued*

electroporated with IRS2-eGFP (WT or S306/577A) and TagRFP-T-Akt2. Cells were stimulated with 1 nM insulin and recruitment assessed by TIRFM (79 WT cells [*n* = 3], 74 S306/577A cells [*n* = 3]). (H) 3T3-L1 adipocytes were co-electroporated with IRS2-eGFP (WT or S306/577A) and phosphoinositide-dependent protein kinase 1 (PDPK1)-TagRFP-T. Cells were stimulated with 1 nM insulin and recruitment assessed by TIRFM (49 WT cells [*n* = 3], 76 S306/577A cells [*n* = 3]). All data expressed as mean ± SEM; PM, plasma membrane; *p<0.05.

The online version of this article includes the following figure supplement(s) for figure 6:

**Figure supplement 1.** Phosphorylation of a variety of IRS1 residues alter IRS1 and Akt localisation at the plasma membrane.

**Figure supplement 2.** Phosphorylation of a variety of IRS2 residues alter IRS2 and Akt localisation at the plasma membrane.

**Figure supplement 3.** Phosphorylation of IRS2 at S365 and S1149 in vitro and in cells.

**Figure supplement 4.** Phosphorylation of IRS1 at S270 and S527 only subtly alter IRS1 and Akt localisation at the plamsa membrane.

phosphorylation of IRS2 S306 and S577 synergistically dissociates PM-associated IRS2 to limit PIP3 production and Akt activation.

We propose a model where in unstimulated cells, a pool of IRS is localised to the PM via a PH domain-dependent interaction. Upon insulin binding its receptor, Akt is rapidly activated (by the canonical IRS/PI3K pathway) and directly phosphorylates IRS proteins at key regulatory residues such as IRS2 S306 and S577. This results in the translocation of IRS from the PM to the cytosol and depletes the pool available to interact with the IR. Ultimately, these events limit PM-associated PI3K and PIP3 synthesis (*Figure 7*).

## Discussion

The PI3K/Akt signalling network is critical to the survival of all eukaryotic cells, and as such the consequences of its dysregulation are severe (*Hers et al., 2011*; *Manning and Toker, 2017*). Aberrant activation of PI3K/Akt signalling underlies a variety of complex diseases, such as cancer and type 2 diabetes (*Hers et al., 2011*; *Manning and Toker, 2017*). Consequently, tight regulation of the PI3K/Akt pathway is critical. Here, we describe an acutely engaged, powerful negative feedback signal that emanates from Akt to IRS1/2 and limits signal flow in a broad range of cell types. Loss of this feedback results in a profound increase in PIP3 production by PI3K. As well as Akt, this feedback has substantial impacts on other PIP3-dependent proteins such as PDPK1 and GAB2. Beyond increasing

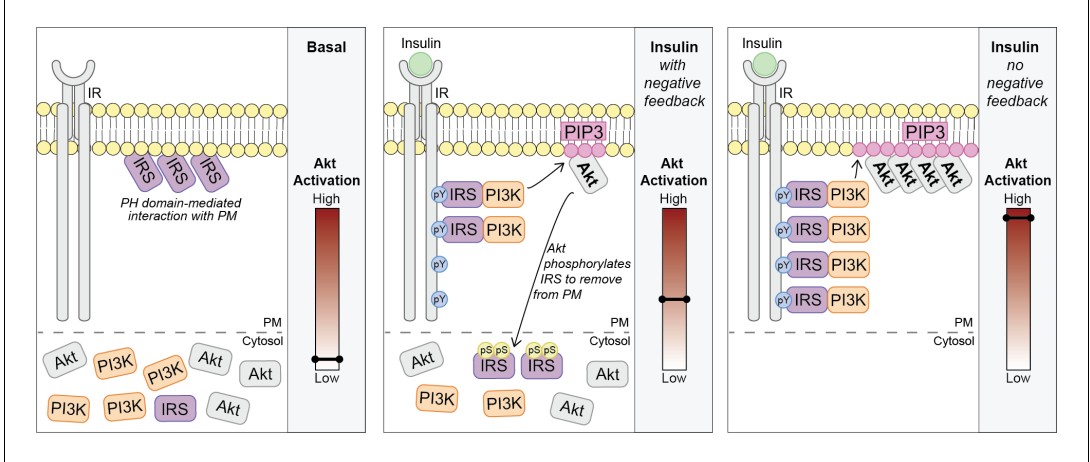

**Figure 7.** Akt phosphorylates insulin receptor substrate 1/2 (IRS1/2) to limit phosphoinositide 3-kinase (PI3K)-mediated phosphatidylinositol (3,4,5)-trisphosphate (PIP3) synthesis. Schematic for a model of Akt-mediated negative feedback. Basal; IRS is plasma membrane (PM) localised due to a pleckstrin homology (PH) domain-mediated interaction with the PM. Insulin (with negative feedback); following insulin binding the insulin receptor (IR), it is autophosphorylated at tyrosine residues (pY) and recruits IRS, which binds PI3K. Plasma membrane-associated PI3K produces PIP3, and Akt is activated. Akt phosphorylates IRS at a number of residues including IRS2 S306 and S577 (pS), which moves IRS to the cytosol and limits the pool available to interact with the IR. This restricts the ability of PI3K to move to the PM and synthesise PIP3. Insulin (no negative feedback); IRS and PI3K are not released from the plasma membrane into the cytosol and consequently there is increased PI3K-mediated PIP3 production and increased Akt activation.

our basic understanding of this crucial signalling node, this discovery has important implications, particularly in cancer therapy, where Akt has long been a major drug target.

Here, we implicate six phosphorylation sites in IRS1 and five phosphorylation sites in IRS2, which are within an Akt substrate motif (R-X-R-X-X-S/T), as major contributors of Akt-mediated negative feedback. In particular, we identify IRS2 as a novel substrate of Akt, and show Akt-mediated phosphorylation of IRS2 at S306 (S303 in mouse) and S577 (S573 in mouse) are primary drivers of negative feedback. We have shown that the phosphorylation of IRS removes it from the PM by disrupting its receptor-independent interactions (*Figure 5I*), and there are several possibilities as to how this occurs. The negative charge of these modifications may repel IRS from the negative electrostatic surface charge of the PM. This has been shown to be the case for other proteins such as MARCKS (*Goldenberg and Steinberg, 2010*). Alternatively, phosphorylation of IRS may promote 14-3-3 binding and sequester IRS away from an interacting protein/lipid at the PM. 14-3-3 has been shown to accompany the movement of the IRS/PI3K complex from membranes to the cytosol (*Xiang et al., 2002*). Intriguingly, IRS2 phosphorylated at S577 has been shown to bind 14-3-3 (*Neukamm et al., 2012*). We suspect that by one of these mechanisms IRS phosphorylation depletes the PM localised pool of IRS which is available to interact with the IR. In addition to negative feedback, it is possible that this movement of IRS to the cytosol might facilitate translocation of the IRS signalling complex to another site, which could be important for signal propagation.

Here, we have shown that the phosphorylation of two IRS2 serine residues by Akt act in synergy to induce IRS2 translocation to the cytosol (*Figure 6G*). We have also shown that there is further synergy between the IRS1/2 phosphorylation sites within an R-X-R-X-X-S/T motif (*Figure 6C–D*). It is possible that this cooperation extends further to IRS1/2 serine/threonine residues phosphorylated by other kinases. A variety of other kinases, including insulin-independent kinases, have been shown to phosphorylate IRS such as JNK, ERK1/2, PKCs, S6K, and mTORC1 (*Copps and White, 2012*). Phosphosite Plus (*Hornbeck et al., 2015*) reports 110 serine/threonine phosphorylation sites on human IRS1 and 132 serine/threonine phosphorylation sites on human IRS2, almost all of which are located within the unstructured tail of IRS1/2. Consequently, IRS phosphorylation on distinct S/T residues could serve as a site of crosstalk between distinct pathways/various kinases, integrating their activation status and modifying the strength of insulin signalling accordingly. Intriguingly, metabolic stress has previously been shown to increase IRS1 serine/threonine phosphorylation and modulate IRS1 tyrosine phosphorylation (*Hançer et al., 2014*). Thus, it is possible that phosphorylation of these other sites could also be a means to induce IRS translocation to the cytosol and impair insulin signalling. It is important to note, however, that modulation of IRS serine/threonine phosphorylation is likely not the mechanism of insulin resistance (*Fazakerley et al., 2019*; *Hoehn et al., 2008*).

We envisage that the described feedback mechanism may have co-evolved with IRS proteins as an obligate intermediate between the IR and PI3K to enhance the signalling capacity of the IR in several ways. First, it may provide the basis for more precise temporal control of insulin signalling. This may be pertinent in that insulin's principal role as a metabolic regulator occurs over a timescale of minutes, while the control of other biological processes such as proliferation or differentiation may not require such fine control as they have different temporal demands. Second, it provides means to specifically regulate insulin signalling whilst keeping PI3K activation by other signalling pathways intact. Finally, as IRS1/2 can be phosphorylated by a range of other kinases (*Copps and White, 2012*), it is possible that the described mechanism transcends to other kinases and enables crosstalk between different growth factor pathways.

Previous studies have identified mTORC1/S6K-dependent feedback signals that regulate Akt activity (*Carlson et al., 2004*; *Copps and White, 2012*; *Harrington et al., 2004*; *Shah et al., 2004*; *Shah and Hunter, 2006*; *Tremblay and Marette, 2001*). However, the feedback signal reported here does not need mTORC1 activation; rapamycin had no effect on acute Akt recruitment, phosphorylation (*Figure 1G,H*), or IRS1/2 localisation (*Figure 5C,D*). Rather, our data suggest an Akt-derived feedback signal that is activated acutely (~1 min) following growth factor exposure. We suspect that described negative feedbacks emanating from mTORC1/S6K are much slower than the negative feedback described in this study, particularly those that rely on the degradation of IRS.

Recent findings by our lab and others have shown that in the absence of S474 phosphorylation, Akt is active and sustains T309 phosphorylation (*Beg et al., 2017*; *Jacinto et al., 2006*; *Kearney et al., 2019*). Indeed, here we show that negative feedback is engaged in the absence of S474 phosphorylation (*Figure 2I*). Correspondingly, we have previously indicated a role for Akt

phosphorylated at T309 alone in driving a positive feedback loop to activate mTORC2 and enhance S474 phosphorylation (*Yang et al., 2015*). Collectively these findings demonstrate that Akt phosphorylated at T309, but not S474 is capable of substrate phosphorylation and sufficient for eliciting feedback mechanisms.

Losing negative feedback from Akt to IRS resulted in a profound increase in PIP3 levels (*Figure 3A*). The catastrophic consequences of such an increase in signal flow is exemplified in cancer. Hyperactivation of Akt due to upstream genetic lesions (e.g., PI3K, PTEN) or mutation in Akt itself (e.g., E17K) can result in uncontrolled regulation of processes such as cell proliferation and survival (*Altomare and Testa, 2005*). As Akt is frequently hyperactivated in human cancers, it has been the target of numerous cancer therapeutics. However, despite the perceived potential, no Akt inhibitors have been approved for use (*Nitulescu et al., 2018*). Akt inhibitors such as MK2206 and GDC0068 have been tested in clinical trials; however, severe side effects were experienced and substantial tumour shrinkage was generally not observed (*Saura et al., 2017*; *Xing et al., 2019*; *Yap et al., 2011*). The data presented herein demonstrate that the therapeutic targeting of kinases such as Akt can have unexpected effects on other signalling networks, resulting from loss of feedback and crosstalk. For example, inhibition of Akt increased insulin-stimulated PIP3 levels by more than fivefold (*Figure 3A*) and enhanced PM recruitment of PH domain containing proteins PDPK1 and GAB2 (*Figure 3B–D*). PDPK1 is the master regulator of other kinases such as PKC, S6K, and SGK, and so PDPK1 mislocalisation at the PM would likely influence these networks. Thus, the central importance of PIP3 beyond Akt signalling (*Czech, 2000*) may have contributed to the lack of efficacy Akt inhibitors have had in the clinic. These data highlight the need for a more detailed understanding of feedback and crosstalk across signalling networks in order to generate effective therapeutics.

# Materials and methods

## Key resources table

| Reagent type (species) or resource | Designation | Source or reference | Identifiers | Additional information |
|---|---|---|---|---|
| Cell line (mouse) | 3T3-L1 | Dr Howard Green, Harvard Medical School | RRID:CVCL_0A20 | |
| Cell line (human) | HEK293E | American Type Culture Collection | RRID:CVCL_6974 | |
| Cell line (human) | HeLa | American Type Culture Collection | RRID:CVCL_0030 | |
| Cell line (human) | HCC1937 | American Type Culture Collection | RRID:CVCL_0290 | |
| Cell line (human) | MCF7 | Associate Prof. Jeff Holst, Centenary Institute | RRID:CVCL_0031 | |
| Antibody | pAkt T309 (rabbit polyclonal) | Cell Signalling Technology | Cat#9275 | WB (1:1000) |
| Antibody | pAkt T309 (rabbit monoclonal) | Cell Signalling Technology | Cat#13038 | WB (1:1000) |
| Antibody | pAkt S474 (mouse monoclonal) | Cell Signalling Technology | Cat#4051 | WB (1:1000) |
| Antibody | pAS160 T642 (rabbit polyclonal) | Cell Signalling Technology | Cat#4288 | WB (1:1000) |
| Antibody | pPRAS40 T246 (rabbit monoclonal) | Cell Signalling Technology | Cat#2997 | WB (1:1000) |
| Antibody | pGSK3α/β S21/29 (rabbit monoclonal) | Cell Signalling Technology | Cat#9327 | WB (1:1000) |
| Antibody | pFOXO1 S256 (rabbit polyclonal) | Cell Signalling Technology | Cat#9461 | WB (1:1000) |
| Antibody | 14-3-3 (rabbit polyclonal) | Santa Cruz | Cat#sc-629 | WB (1:1000) |
| Antibody | pP70S6K T389 (rabbit polyclonal) | Cell Signalling Technology | Cat#9205 | WB (1:1000) |

*Continued on next page*

Continued

| Reagent type (species) or resource | Designation | Source or reference | Identifiers | Additional information |
|---|---|---|---|---|
| Antibody | Akt2 (rabbit monoclonal) | Cell Signalling Technology | Cat#3063 | WB (1:1000) |
| Antibody | Caveolin (mouse monoclonal) | Abcam | Cat#ab17052 | WB (1:1000) |
| Antibody | Pan Akt (rabbit monoclonal) | Cell Signalling Technology | Cat#4685 | WB (1:1000) |
| Antibody | PI3K p85 (rabbit monoclonal) | Cell Signalling Technology | Cat#4257 | WB (1:1000) IF (1:100) |
| Antibody | PI3K p110 (rabbit monoclonal) | Cell Signalling Technology | Cat#4249 | WB (1:1000) |
| Antibody | IRS1 (rabbit polyclonal) | Cell Signalling Technology | Cat#2382 | WB (1:1000) |
| Antibody | Na,K-ATPase (rabbit polyclonal) | Prof. Gus Lienhard | | |
| Antibody | IR (rabbit monoclonal) | Cell Signalling Technology | Cat#3025 | WB (1:1000) |
| Antibody | pTyrosine (rabbit monoclonal mix) | Cell Signalling Technology | Cat#8954 | WB (1:1000) |
| Antibody | α-Tubulin (mouse monoclonal) | Sigma-Aldrich | Cat#T9026 | WB (1:1000) |
| Antibody | IRS1 (rabbit monoclonal) | Cell Signalling Technology | Cat#3407 | IP (2 µL per sample) |
| Antibody | IRS2 (rabbit polyclonal) | Cell Signalling Technology | Cat#3089 | IP (2 µL per sample) |
| Antibody | FLAG (mouse monoclonal) | Sigma-Aldrich | Cat#F1804 | IP (5 µg per sample) |
| Antibody | IgG control (rabbit) | Santa Cruz | Cat#sc-2027 | IP (control) |
| Antibody | Alexa Fluor 555 (goat polyclonal) | Thermo Fisher Scientific | Cat#A21428 | IF(1:200) |
| Peptide, recombinant protein | 3X FLAG Peptide | Sigma-Aldrich | Cat#F4799 | |
| Chemical compound, drug | Insulin | Sigma-Aldrich | Cat# I5500; CAS 11070-73-8 | |
| Chemical compound, drug | Insulin-like growth factor 1 (IGF1) | Miltenyi Biotec | Cat#130-093-886 | |
| Chemical compound, drug | Epidermal growth factor (EGF) | Miltenyi Biotec | Cat#130-097-749 | |
| Chemical compound, drug | GDC0068 | Selleck Chemicals | Cat#S2808; CAS 1001264-89-6 | |
| Chemical compound, drug | MK2206 | MedChemExpress | Cat#HY-10358; CAS 1032350-13-2 | |
| Chemical compound, drug | Wortmannin | Sigma-Aldrich | Cat#W1628; CAS 19545-26-7 | |
| Chemical compound, drug | Rapamycin | LC Laboratories | Cat#R-5000; CAS 53123-88-9 | |
| Chemical compound, drug | Rapalink | Prof. Kevan Shokat (*Rodrik-Outmezguine et al., 2016*) | | |
| Chemical compound, drug | GSK2334470 | Tocris | Cat# 4143; CAS 1227911-45-6 | |
| Commercial assay or kit | PIP3 ELISA | Echelon Biosciences | Cat#K-2500S | |
| Commercial assay or kit | PI(3,4)P2 ELISA | Echelon Biosciences | Cat#K-4500 | |
| Recombinant DNA reagent | TagRFP-T-Akt2 (plasmid) | *Norris et al., 2017* | | |

*Continued*

| Reagent type (species) or resource | Designation | Source or reference | Identifiers | Additional information |
|---|---|---|---|---|
| Recombinant DNA reagent | TagRFP-T-Akt2 W80A; W80A-T309A; W80A-S474A; W80A-T309A-S474A; W80A-K181A (plasmids) | This paper | | See Materials and methods and *Supplementary file 3* for DNA sequences |
| Recombinant DNA reagent | PDPK1-eGFP (plasmid) | This paper | | See Materials and methods and *Supplementary file 3* for DNA sequences |
| Recombinant DNA reagent | PDPK1-TagRFP-T (plasmid) | This paper | | See Materials and methods and *Supplementary file 3* for DNA sequences |
| Recombinant DNA reagent | PH-Gab2-GFP (plasmid) | Sergio Grinstein – Addgene | Plasmid #35147 | |
| Recombinant DNA reagent | PI3K p110* (plasmid) | Morris Birnbaum | | |
| Recombinant DNA reagent | pMIG FLAG-Akt2 W80A; W80A-T309A (plasmids) | *Kearney et al., 2019* | | |
| Recombinant DNA reagent | IRS1-eGFP (plasmid) | This paper | | See Materials and methods and *Supplementary file 3* for DNA sequences |
| Recombinant DNA reagent | IRS1-eGFP S270A; S307A; S330A; T525A; S527A; S1101A; S270A-S307A-S330A-T525A-S527A-S1101A (6P); S270A-S527A; DelPH; DelPTB (plasmids) | This paper | | See Materials and methods and *Supplementary file 3* for DNA sequences |
| Recombinant DNA reagent | IRS2-eGFP (plasmid) | This paper | | See Materials and methods and *Supplementary file 3* for DNA sequences |
| Recombinant DNA reagent | IRS2-eGFP S306A; S346A; S365A; S577A; S1149A; S306A-S346A-S365A-S577A-S1149A (5P); S306A-S577A (plasmids) | This paper | | See Materials and methods and *Supplementary file 3* for DNA sequences |
| Recombinant DNA reagent | IRS2-FLAG (plasmid) | This paper | | See Materials and methods and *Supplementary file 3* for DNA sequences |
| Sequence-based reagent | DNA Primers | Sigma-Aldrich | | See Materials and methods and *Supplementary file 3* for DNA sequences |
| Software, algorithm | MaxQuant | *Cox and Mann, 2008*; *Tyanova et al., 2016* | RRID:SCR_014485 | |
| Software, algorithm | Image Studio | LI-COR Biosciences | RRID:SCR_013715 | |
| Software, algorithm | Graphpad Prism | GraphPad Software Inc. | RRID:SCR_002798 | |
| Software, algorithm | Fiji ImageJ | *Schindelin et al., 2012* | RRID:SCR_003070 | |
| Software, algorithm | Tableau Prep | Tableau Software | | |
| Software, algorithm | R | https://www.R-project.org/ | RRID:SCR_001905 | |
| Software, algorithm | MATLAB | MathWorks | RRID:SCR_001622 | |
| Software, algorithm | IQM | IntiQuan | | |

## Modelling

We constructed nine mechanistic models to investigate different possible network structures of the PI3K/Akt signalling pathway. All models included components representing proximal insulin signalling proteins (IR, IRS, PI3K). The activation of this pathway is initiated by insulin binding the IR. Then, IRS bind the IR and recruit PI3K to the PM. In the model, the IRS/PI3K node represents proximal insulin signalling. Activated PI3K phosphorylates PIP2, converting it to PIP3. This is negatively regulated by PTEN. In the model, Akt is activated by PDPK1 and mTORC2 through phosphorylation at

T309 and S474, respectively. mTORC2 has two independent activation mechanisms: (1) binding of SIN1 to PIP3 that releases its inhibition on mTOR kinase activity (*Liu et al., 2015*) and (2) phosphory-lation of SIN1 T86 residue by Akt (*Humphrey et al., 2013*; *Yang et al., 2015*). Activated Akt phos-phorylates PRAS40, which results in mTORC1 activation. In the model, Akt singly phosphorylated at T309, and doubly phosphorylated at T309 and S474 has kinase activity. However, Akt singly phos-phorylated at S474 is not active. All models were trained with the data in *Figure 1B and C,D* (pAkt T309, pAkt S474, and pPRAS40 T246).

Additionally, model 1 included negative feedback from mTORC1 to IRS/PI3K. Model 2 included negative feedback from mTORC1 to IRS/PI3K and was also trained with the data in *Figure 1G*. Model 3 incorporated no negative feedback. Model 4 included negative feedback from PDPK1 to IRS/PI3K. Model 5 included negative feedback from PDPK1 to PTEN. Model 6 included negative feedback from mTORC2 to IRS/PI3K. Model 7 included negative feedback from mTORC2 to PTEN. Model 8 included negative feedback from Akt to PTEN. Model 9 included negative feedback from Akt to IRS/PI3K. Detailed schematic diagrams of these models are illustrated in *Figures 1A* and *2A*.

These models were formulated using ODEs. The rate equations, ODEs, and the best-fitted parameter sets for each network model are given in *Supplementary file 1*. The code for the model-ling has been deposited to Github (*Ghomlaghi et al., 2021*). The model construction and calibration processes were implemented in MATLAB (The MathWorks Inc 2019a) and the IQM toolbox (http://www.intiquan.com/intiquan-tools/) was used to compile the IQM file for a MEX file which makes the simulation much faster.

The quality of a mathematical model is generally justified by its ability to recapitulate known experimental data. Model calibration (or model training) is the process of estimation of the model's parameters. This process produces a 'best-fitted' model that best recapitulates biological observa-tions used for model calibration. Model calibration was done by estimating the model parameter val-ues to minimise an objective function $J$ that quantifies the difference between model simulation results and corresponding experimental measurements:

$$J(p) = \sum_{j=1}^{M} w_j \sum_{i=1}^{N} \left( \frac{y_{j,i}^D - y_j(t_i, p)}{\sigma_{j,i}} \right)^2$$

Here, $M$ denotes number of available experimental data sets for fitting and $N$ is the number of time points in each experimental data set. $y_j(t_i, p)$ is simulation result of the model for the component $j$ in the network at the time point $t_i$ while parameter set $p$ is used for the simulation. Finally, $y_{j,i}^D$ is the mean value of the experimental data of component $j$ at time point $t_i$ with the error variance $\sigma_{j,i}$. $w_j$ is the weight of the component $j$.

A genetic algorithm (GA) was used to optimise the objective function (*Man et al., 1996*; *Reali et al., 2017*; *Shin et al., 2014*). This was done by using the Global Optimization Toolbox and the function *ga* in MATLAB. Selection rules in GA select the individual solutions with the best fitness values (called 'elite solutions') from the current population. The elite count was set to 5% of the pop-ulation size. Crossover rules combine two parents to generate offspring for the next generation. The crossover faction was set at 0.8. Mutation rules apply random changes to individual parents to gen-erate the population of the next generation. For the mutation rule, we generated a random number from a Gaussian distribution with mean 0 and standard deviation $\sigma_k$, which was applied to the indi-viduals of the current generation. The standard deviation function ($\sigma_k$) is given by the recursive for-mula as follows:

$$\sigma_k = \sigma_{k-1} \left( 1 - \frac{k}{G} \right)$$

where $k$ is the $k$th generation, $G$ is the number of generation, and $\sigma_0 = 1$. To derive the best-fitted parameter sets, we carried out repeated GA runs with population size of 200 and the generation number set to 800.

## Modelling identifiability analysis

Identifiability analysis for all tested models (models 4–9; *Figure 2A*) was performed using a well-established method based on profile likelihood (*Maiwald et al., 2016*; *Rateitschak et al., 2012*;

*Raue et al., 2009*). This method is able to detect both structural and practical non-identifiable parameters through calculating confidence intervals defined by a threshold in the profile likelihoods, as detailed previously (*Rateitschak et al., 2012*). A parameter is considered to be identifiable if the confidence interval of its estimate is finite (*Rateitschak et al., 2012*). Another important advantage of this approach is that it is computationally efficient and thus suitable for medium-to-large (non-minimal) models such as those considered in this paper (*Rateitschak et al., 2012*).

For normally distributed observational noise, this function corresponds to the maximum likelihood estimate of $\theta$. The profile likelihood of a parameter $\theta$ is given by *Maiwald et al., 2016*; *Rateitschak et al., 2012*; *Raue et al., 2009*:

$$\chi^2_{PL}(\theta_i) = \min_{\theta_{j \neq i}} \chi^2(\theta)$$

which represents a function in $\theta_i$ of least increase in the residual sum of squares $\chi(\theta)$.

Profile likelihood-based confidence interval (CI) can be derived via:

$$CI(\theta) = \left\{ \theta | \chi^2_{PL}(\theta) - \chi^2_{PL}\left(\hat{\theta}\right) < \Delta_\alpha \right\}$$

where $\Delta_\alpha = \chi^2(\alpha, df)$ is the threshold, $\alpha$ is a confidence level (the $\alpha$ quantile of the $\chi^2$ distribution), $df$ is the degree of freedom ($df = 1$ for pointwise confidence interval and $df = \#\,of\,parameters$ for simultaneous confidence intervals, respectively). $\theta$ denotes the best-fitted parameter set.

*Supplementary file 2* contains the identifiability analysis results for models 4–9. In each plot, the black dashed lines depict pointwise confidence levels of $\alpha = 95\%$. The results show for each parameter whether it is structurally (indicated by a flat curve in both directions) or practically (indicated by a flat curve only in one direction) non-identifiable (*Rateitschak et al., 2012*). Overall, these results demonstrate that all the models are non-identifiable, meaning each model has at least one non-identifiable parameter. As most large ODE-based models in systems biology are unidentifiable to some extent, the results here were expected given the detailed scope of our models, which were designed to capture the important biological mechanisms within the insulin signalling network, including multiple feedback/feedforward loops. Because generally there is a trade-off between model identifiability and level of biological details, although the models in this study could be made more identifiable through model abstraction, such a process is inevitably at the cost of sacrificing specific biological details and may weaken the models' explanatory and predictive power.

## Cloning

TagRFP-T-Akt2 consists of human Akt2 tagged with TagRFP-T at its N-terminus as described previously (*Norris et al., 2017*). TagRFP-T-Akt2 W80A, W80A-T309A, W80A-S474A, W80A-T309A-S474A, and W80A-K181A were generated using site-directed mutagenesis (*Sanchis et al., 2008*). PDPK1-eGFP consists of human PDPK1 tagged with eGFP at its C-terminus. To generate PDPK1-eGFP, R777-E159 Hs.PDPK1 was kindly gifted from Dominic Esposito (Addgene plasmid #70443) and was used as a template to amplify human PDPK1. Human PDPK1 was placed in the pEGFP-C1 vector using Gibson assembly cloning (*Gibson, 2011*). PH-Gab2-GFP was a gift from Sergio Grinstein (Addgene plasmid #35147). Constitutively active p110 (p110*) was kindly gifted by Morris Birnbaum. pMIG FLAG-W80A and FLAG-W80A-T309A Akt2 were generated as previously described (*Kearney et al., 2019*). IRS1-eGFP consists of human IRS1 tagged with eGFP at its C-terminus. To generate IRS1-eGFP, R777-E109 Hs.IRS1 was kindly gifted from Dominic Esposito (Addgene plasmid #70393) and was used as a template to amplify human IRS1. Human IRS1 was placed in the pEGFP-C1 vector using Gibson assembly cloning (*Gibson, 2011*). IRS2-eGFP consists of human IRS2 tagged with eGFP at its C-terminus. To generate IRS2-eGFP, R777-E111 Hs.IRS2 was kindly gifted from Dominic Esposito (Addgene plasmid #70395) and was used as a template to amplify human IRS2. Human IRS2 was placed in the pEGFP-C1 vector using Gibson assembly cloning (*Gibson, 2011*). Mutations in IRS1-eGFP and IRS2-eGFP were generated using site-directed mutagenesis (*Sanchis et al., 2008*). DelPH IRS1-eGFP consists of IRS1 without its PH domain (residues 2–115 removed, based on *Dhe-Paganon et al., 1999*) and was generated using Gibson assembly cloning (*Gibson, 2011*). DelPTB IRS1-eGFP consists of IRS1 without its PTB domain (residues 161–265 removed, based on *Eck et al., 1996*) and was generated using Gibson assembly cloning (*Gibson, 2011*). To generate IRS2-FLAG, IRS2-eGFP was used as a template, and eGFP replaced with a

FLAG tag (DYKDDDDK) using Gibson assembly cloning (*Gibson, 2011*). To generate PDPK1-TagRFP-T, TagRFP-T-Akt2 was used as a template to amplify TagRFP-T, and this was inserted into PDPK1-eGFP to replace eGFP, using Gibson assembly cloning (*Gibson, 2011*). Plasmid and primer DNA sequences are provided in *Supplementary file 3*.

## Cell culture

3T3-L1 fibroblasts obtained from the Howard Green Laboratory (Harvard Medical School) were cultured in high glucose Dulbecco's modified eagle medium (DMEM) (Gibco by Life Technologies) supplemented with 10% (v/v) fetal bovine serum (FBS) (Gibco by Life Technologies), and 1× GlutaMAX (Gibco by Life Technologies) at 37°C and 10% $CO_2$. Cells were differentiated into adipocytes as described previously (*Fazakerley et al., 2015*; *Norris et al., 2018*) and used for experiments 7–12 days after initiation of differentiation. 3T3-L1 adipocytes stably expressing FLAG-W80A or FLAG-W80A-T309A Akt2 were generated using retrovirus as previously described and characterised previously (*Kearney et al., 2019*). HEK293E, HeLa, and HCC1937 cell lines were obtained from the American Type Culture Collection and grown in the medium described above to culture 3T3-L1 cells. MCF7 cells were a gift from Associate Prof. Jeff Holst (Centenary Institute) and were validated by STR. Cells were maintained in Minimum Essential Media (Gibco by Life Technologies Cat#10370–021), with the addition of 10% (v/v) FBS, 1× GlutaMAX, and 1 mM sodium pyruvate (Gibco by Life Technologies) at 37°C and 5% $CO_2$. Cells were routinely tested for mycoplasma contamination and found to be contamination-free.

## Western blotting

3T3-L1 adipocytes were serum-starved with DMEM containing 1× GlutaMAX and 0.2% BSA (w/v) for 2 hr. Cells were then exposed to drugs/insulin. Cells were then placed on ice, washed with cold PBS, lysed with 1% (w/v) SDS in PBS containing protease inhibitors (Roche Applied Science) and phosphatase inhibitors (2 mM $Na_3VO_4$, 1 mM $Na_4O_7P_2$, and 10 mM NaF), and tip probe-sonicated. Lysates were centrifuged at 13,000× *g* for 15 min at 4°C. The lipid layer was removed, and protein content was quantified using the Pierce BCA Protein Assay Kit (Thermo Scientific); 10 μg of lysate was then resolved by SDS-PAGE and transferred to PVDF membranes. Membranes were blocked and immunoblotted as described previously (*Fazakerley et al., 2015*). Densitometry analysis was performed using ImageStudioLite version 5.2.5 (LI-COR). Band intensities were normalised to the loading control. Statistical tests were performed using GraphPad Prism version 7.0. For the quantification of the blots in *Figure 1C,D*, the time courses were normalised to the mean intensity of all samples (within a blot). Next, biological replicates were normalised to the maximum of the mean of all responses (across blots) within a dose. As some of the 1 and 100 nM time courses were acquired separately, the difference in magnitude between the doses was determined by the three biological 1 and 100 nM replicates that were acquired concurrently and run on the same gels. The representative blot is an example of a paired experiment.

## Live cell TIRFM

3T3-L1 adipocytes were electroporated 6–8 days post-differentiation with 6–10 μg of plasmid and placed onto the Matrigel-coated μ-Dish 35 mm, high Glass Bottom coverslips (Ibidi) as described previously (*Norris et al., 2017*). For other cell types, cells were transfected using Lipofectamine 2000 (Thermo Scientific). Twenty-four hours later, cells were serum-starved for 2 hr and then incubated at 37°C with Krebs-Ringer-phosphate-HEPES buffer (0.6 mM $Na_2HPO_4$, 0.4 mM $NaH_2PO_4$, 120 mM NaCl, 6 mM KCl, 1 mM $CaCl_2$, 1.2 mM $MgSO_4$, and 12.5 mM HEPES [pH 7.4]) supplemented with 10 mM glucose, 1× minimum essential medium amino acids (Gibco by Life Technologies), 1× GlutaMAX, and 0.2% (w/v) BSA. While imaging, temperature and humidity were then maintained using an Okolab cage incubator and temperature control. The cells were treated using a custom-made perfusion system. Images were acquired with a CFI Apochromat TIRF 60× oil, NA 1.49 objective, using the Nikon Ti-LAPP H-TIRF module angled to image ~90 nm into cells. Images were acquired approximately every 15 s. To quantify changes in the PM recruitment of each protein of interest, we measured the average pixel intensity (and subtracted background intensity) for each cell over the time course using Fiji (*Schindelin et al., 2012*). Each cellular response to stimuli was normalised to its average intensity over the basal period. The cell-to-cell heterogeneity in Akt recruitment

responses (described previously; *Norris et al., 2021*) can make the comparison of several large population TIRF responses difficult to interpret if presented as mean ± SD. These data are presented as mean ± SEM to aid interpretation. Rate constants (*Figure 4B–D*) were calculated using Graphpad Prism version 7.0, by fitting an exponential curve to the data (plateau followed by one-phase decay). To assess the relative cell surface level of IRS1 (*Figure 5F*), TIRF and epifluorescence images were acquired and for each cell its median TIRF intensity (corrected for background) was normalised to its median epifluorescence intensity (corrected for background) using Fiji (*Schindelin et al., 2012*).

## PI(3,4,5)P3 and PI(4,5)P2 Mass ELISA

3T3-L1 adipocytes were serum-starved with DMEM containing 1× GlutaMAX and 0.2% (w/v) BSA for 2 hr. Cells were then preincubated with drugs or vehicle controls and stimulated with 1 nM insulin. Lipids were extracted from cells and measured using an ELISA kit (Echelon Biosciences). 1 × 10 cm dish of 3T3-L1 adipocytes was used for each sample. For each sample, PIP3 mass was normalised to PIP(4,5)P2 mass. PIP(4,5)P2 is highly abundant in cells (*Guillou et al., 2007*) and thus is only marginally affected by changes in PI3K activity (*Condliffe et al., 2005*). As has been done previously (*Clark et al., 2011*; *Costa et al., 2015*; *Guillou et al., 2007*), we normalised to PI(4,5)P2 mass to account for differences in extraction efficiency between samples, and control for total cellular phosphoinositides.

## Subcellular fractionation

3T3-L1 adipocytes were serum-starved for 2 hr, and then exposed to DMSO, 10 µM MK2206, or 10 µM GDC0068 for 5 min, followed by 1 nM insulin for 10 min. Cells were placed on ice, washed with cold PBS, and harvested in cold HES buffer (20 mM HEPES, 1 mM EDTA, 250 mM sucrose, pH 7.4) containing phosphatase (2 mM $Na_3VO_4$, 1 mM $Na_4O_7P_2$, 10 mM NaF) and protease (Roche Applied Science) inhibitors. All subsequent steps were carried out at 4°C. Cells were homogenised by passing through a 22-gauge needle 10 times and a 27-gauge needle six times prior to centrifugation at 500× *g* for 10 min. The supernatant was centrifuged at 13,550× *g* for 12 min to pellet the PM and mitochondria/nuclei, while the supernatant contained the cytosol, LDM fraction, and HDM fraction. The supernatant was then centrifuged at 21,170× *g* for 17 min to pellet the HDM fraction. That supernatant was then centrifuged at 235,200× *g* for 75 min to obtain the cytosol fraction (supernatant) and the LDM fraction (pellet). The PM and mitochondria/nuclei pellet was resuspended in HES buffer and again centrifuged at 13,550× *g* for 12 min. The pellet was then resuspended in HES buffer, layered over high sucrose HES buffer (20 mM HEPES, 1 mM EDTA, 1.12 M sucrose, pH 7.4), and centrifuged at 111,160× *g* for 60 min in a swing-out rotor. The PM fraction was collected at the interface between the sucrose layers, and pelleted by centrifugation at 235,200× *g* for 75 min. All pellets were resuspended in HES buffer containing phosphatase and protease inhibitors. Protein concentrations for each fraction were determined using the Pierce BCA Protein Assay Kit (Thermo Scientific).

## Immunofluorescence/TIRFM

3T3-L1 adipocytes were seeded onto Matrigel-coated eight-well microslides (Ibidi). Forty-eight hours later, cells were serum-starved for 2 hr and then exposed to DMSO, 10 µM MK2206, or 10 µM GDC0068 for 5 min, followed by 1 nM insulin for 10 min. The coverslips were then briefly immersed in ice-cold PBS and fixed with 4% paraformaldehyde in PBS at room temperature for 15 min. Cells were then washed twice with room temperature PBS and quenched with 200 mM glycine for 10 min. Cells were then blocked and permeabilised with 2% BSA/0.1% saponin in PBS for 30 min. Cells were incubated with the anti-PI3K p85 (Cell Signaling Technology CST4257S) primary antibody (1:100 in 2% BSA/0.1% saponin in PBS) for 1 hr at room temperature. Cells were then washed with 2% BSA/0.1% saponin in PBS three times, and then incubated with anti-rabbit-IgG conjugated to Alexa Fluor 555 (1:200 in 2% BSA/0.1% saponin in PBS) at room temperature for 1 hr in the dark. Cells were then washed five more with PBS and then stored and imaged in 5% glycerol/2.5% 1,4-diazabicyclo[2.2.2]octane in PBS. Images were acquired using the Nikon Ti-LAPP H-TIRF module angled to image ~90 nm into cells. To quantify relative changes in PM PI3K p85, for each cell we measured the median pixel intensity (corrected for background) using Fiji (*Schindelin et al., 2012*).

## Endogenous IRS1 and IRS2 immunoprecipitation for LC-MS/MS

3T3-L1 adipocytes were serum-starved for 2 hr and then exposed to DMSO, 10 µM MK2206, or 10 µM GDC0068 for 5 min, followed by 1 nM insulin for 10 min. Cells were washed three times with ice-cold PBS and lysed in cold lysis buffer (1% (v/v) NP40, 10% (v/v) glycerol, 137 mM NaCl, 25 mM Tris pH 7.4) containing phosphatase (2 mM $Na_3VO_4$, 1 mM $Na_4O_7P_2$, 10 mM NaF) and protease (Roche Applied Science) inhibitors. All subsequent steps were performed at 4°C. Lysates were passed through a 22-gauge needle 10 times, followed by a 27-gauge needle six times. Lysates were then solubilised for 15 min on ice prior to centrifugation at 18,000× $g$ for 20 min to remove lipid and cell debris; 850 µg of each supernatant was then incubated with 2 µL of antibody (IRS1; Cell Signaling Technology CST3407S or IRS2; Cell Signaling Technology CST3089S) or the same amount of Rabbit IgG control (Santa Cruz) for 2 hr with rotation. 50 µL of Dynabeads Protein G (Invitrogen) were added into each antibody-lysate mixture and incubated for 2 hr with rotation. Beads were washed once with lysis buffer and then four times with PBS. Beads were incubated in 25 µL elution buffer 1 (2 M urea, 5 mM TCEP, 20 mM 2-chloroacetamide, 5 µg/mL trypsin, 50 mM Tris-HCl, pH 7.5) for 30 min at room temperature, and then 100 µL elution buffer 2 (2 M urea, 50 mM Tris-HCl, pH 7.5) was added. Eluate was collected into a LowBind Eppendorf tube and digested for 16 hr at room temperature. Peptides were then acidified by adding TFA to a final concentration of 1% (v/v) and stored at 4°C prior to LC-MS/MS.

## IRS2-FLAG immunoprecipitation and in vitro kinase assay for LC-MS/MS

IRS2-FLAG was transfected into HEK293E cells using Lipofectamine 2000 (Thermo Scientific). Twenty-four hours later, cells were serum-starved for 2 hr and treated with 10 µM MK2206 for 30 min. Cells were placed on ice, washed with cold PBS, and harvested in cold lysis buffer (1% (v/v) NP40, 10% (v/v) glycerol, 137 mM NaCl, 25 mM Tris pH 7.4) containing phosphatase (2 mM $Na_3VO_4$, 1 mM $Na_4O_7P_2$, 10 mM NaF) and protease (Roche Applied Science) inhibitors. All subsequent steps were carried out at 4°C. Cells were homogenised by passing through a 22-gauge needle 10 times and a 27-gauge needle six times prior to solublisation on ice for 15 min. Then samples were centrifuged at 18,000× $g$ for 20 min; 1 mg of the supernatant was incubated with 5 µg of anti-FLAG antibody (Sigma-Aldrich), on a rotator for 2 hr. Then, this was mixed with protein G agarose beads (GE Healthcare) on a rotator for a further 2 hr. Beads were washed four times with lysis buffer, once with kinase buffer (25 mM Tris-HCl [pH 7.5], 10 mM $MgCl_2$, 5 mM beta-glycerophosphate, 0.1 mM $Na_3VO_4$, 1 mM DTT), and then dried. To elute the protein from the beads, 0.4 µg/µL of 3× FLAG peptide (Sigma-Aldrich) in kinase buffer was added to each sample and incubated for 1 hr with rapid agitation (1500 rpm using an Eppendorf ThermoMixer C). The eluate was removed. To determine the concentration of IRS2-FLAG obtained, an aliquot of eluate and Albumin standards (Thermo Scientific) were resolved by SDS-PAGE and stained using SYPRO Ruby Protein Gel Stain (Bio-Rad). Remaining eluate was stored at −20°C for further analysis.

100 ng of immunoprecipitated IRS2-FLAG protein, 30 ng of recombinant active Akt2 (Signal-Chem), and 100 µM ATP were mixed in kinase buffer (25 mM Tris-HCl [pH 7.5], 10 mM $MgCl_2$, 5 mM beta-glycerophosphate, 0.1 mM $Na_3VO_4$, 1 mM DTT). Samples were incubated at 30°C for 1 hr with rapid agitation (500 rpm). Samples were placed at 70°C for 10 min, then cooled on ice to room temperature. Proteins were reduced and alkylated by the addition of 10 mM TCEP (Thermo Scientific, Bond-Breaker TCEP solution, Neutral pH) and 40 mM 2-chloroacetamide (Sigma-Aldrich) and incubated at 45°C for 5 min. Samples were cooled on ice to room temperature, and 1% (w/v) SDC (in 25 mM Tris pH 7.5) added to the samples; 10 ng of trypsin and 10 ng of LysC were added and samples shaken at 37°C with rapid agitation (2000 rpm) for 18 hr. Samples were then mixed with equal volume of 1% TFA in ethyl acetate (45 µL) and vortexed to dissolve precipitated SDC. Peptides were desalted using StageTips (*Rappsilber et al., 2003*) using SDB-RPS solid-phase extraction discs (Empore). Briefly, 200 µL tips were packed with two layers of SDB-RPS material and placed into a 3D-printed 96-well StageTips adapter (*Harney et al., 2019*) for centrifugation. StageTips were equilibrated with sequential 50 µL washes of 100% acetonitrile, 30% MeOH with 1% TFA, and 0.2% TFA in water by centrifugation at 1000× $g$ for 2 min. Peptides were then loaded onto the StageTips by centrifugation at 1000× $g$ for 2 min. StageTips were washed sequentially with 1% TFA in ethyl acetate, 1% TFA in isopropanol and 0.2% TFA in 5% acetonitrile, and eluted into PCR strip tubes with 5% ammonium hydroxide in 60% ACN. Peptides were concentrated to dryness in a vacuum

concentrator at 45°C for 30 min. Peptides were resuspended in MS loading buffer (10 μL 3% ACN/0.1% TFA) prior to LC-MS/MS analysis.

## Pharmacological inhibition experiments and phosphoproteomics for LC-MS/MS

For GDC0068/MK2206 experiments, 3T3-L1 adipocytes were serum-starved for 2 hr and then exposed to vehicle (DMSO), 10 μM MK2206 or 10 μM GDC0068 for 5 min, followed by 1 nM insulin for 10 min. Cells were harvested in ice-cold SDC lysis buffer (4% sodium deoxycholate/100 mM Tris pH 8.5), boiled at 95°C for 5 min, centrifuged at 18,000× g for 15 min, and the layer of fat removed prior to determining protein concentration by BCA assay.

For rapamycin/rapalink experiments, HEK-293E cells were maintained in DMEM, with 4.5 g glucose/L, 2 mM L-GlutaMAX, and 10% FBS. Cells were passaged for six doublings in stable isotope labelling by amino acids in cell culture (SILAC) DMEM containing three different isotopic versions of lysine and arginine supplemented with 10% dialysed FBS, generating 'triple-labelled' SILAC cells (*Ong et al., 2002*). Cells were serum-starved for 4 hr together with either 100 nM rapamycin, 3 nM RapaLink1, or vehicle (DMSO), and then treated with 100 nM insulin for 10 min. Experiments were performed with four biological replicates and label switching. Cells were harvested in ice-cold GdmCl lysis buffer (6 M GdmCl, 100 mM Tris pH 8.8, 10 nM TCEP, 40 mM CAA). Protein concentration was estimated by BCA assay, and SILAC samples mixed accordingly in equal ratios. Samples were processed using the EasyPhos method (*Humphrey et al., 2015a*) and phosphopeptides were resuspended in MS loading buffer (0.3% v/v TFA/2% (v/v) ACN) prior to LC-MS/MS.

## LC-MS/MS and MS data analysis

For endogenous IRS1/2 immunoprecipitation and IRS2-FLAG in vitro kinase assay samples, peptides were analysed using a Dionex HPLC coupled to a Q-Exactive HF-X benchtop Orbitrap mass spectrometer (Thermo Fisher Scientific). Peptides were injected onto an in-house packed 75 μm ID × 40 cm column packed with 1.9 μm C18 (ReproSil Pur C18-AQ, Dr Maisch) and separated by a binary gradient of buffer A (0.1% formic acid) and buffer B (0.1% formic acid/80% ACN). Peptides were separated by a gradient of 5–30% (IP) or 5–40% (in vitro kinase assay) buffer B at a flow rate of 300 or 400 nL/min. Eluting peptides were directly analysed with one full scan (350–1400 m/z, $R = 60,000$). The top 5 (in vitro kinase assay) or 15 (IP) most intense precursors were fragmented with a collision energy of 27% and MS2 spectra collected at a resolution of 15,000.

For phosphoproteomics, phosphopeptides were loaded onto in-house fabricated 40 cm columns with a 75 μM inner diameter, packed with 1.9 μM C18 ReproSil Pur AQ particles (Dr Maisch GmbH) using EASY-nLC 1000 HPLC. Column temperature was maintained at 60°C using a column oven (Sonation, GmbH). Peptides were separated using a binary buffer system comprising 0.1% formic acid (buffer A) and 80% ACN plus 0.1% formic (buffer B), at a flow rate of 350 nL/min, with a gradient of 3–19% or 3–20% buffer B over 60 min or 85 min, respectively (for the GDC0068/MK2206 and rapamycin/rapalink experiments), followed by 19–41% or 20–45% buffer B over 30 or 45 min, resulting in gradients of approximately 1.5 or 2 hr. Peptides were analysed on a Q Exactive HF or HF-X benchtop Orbitrap mass spectrometer (Thermo Fisher Scientific), with one full scan (300–1600 m/z, $R = 60,000$) at a target of 3e6 ions, followed by up to 5 (HF) or 10 (HF-X) data-dependent MS/MS scans with higher-energy collisional dissociation (HCD) fragmentation. MS2 scan settings: target 1e5 ions, max ion fill time 120 ms (HF) or 50 ms (HF-X), isolation window 1.6 m/z, normalised collision energy (NCE) 25% (HF) or 27% (HF-X), intensity threshold 3.3e5 ions (HF) or 4e5 ions (HF-X), with fragments detected in the Orbitrap ($R = 15,000$). Dynamic exclusion (40 s, HF; 30 s HF-X) and apex trigger (4–7 s, HF; 2–4 s HF-X) were enabled.

RAW MS data was analysed using using MaxQuant (*Cox and Mann, 2008*) with searches performed against the UniProt database (June 2020 [in vitro kinase assay], June 2019 [IRS1/2 IP], November 2016 [rapamycin/rapalink phosphoproteome], and March 2018 [GDC0068/MK2206 phosphoproteome] releases) with a false discovery rate of <0.01 at the protein, peptide, site, and PSM levels. Default settings in MaxQuant were used, with the addition of 'Phospho(STY)' as a variable modification and SILAC labels (Arg 0/Lys 0, Arg 6/Lys 4, Arg 10, Lys 8) for the rapamycin/rapalink phosphoproteome samples. 'Match between runs' was enabled with a 0.7 min match time window (default).

Data was filtered, normalised, and analysed using R, Tableau Prep, and Graphpad Prism. For the phosphoproteomics with GDC0068/MK2206, LFQ Intensities were $\log_2$-transformed and median normalised. For endogenous IRS1/2 immunoprecipitation, intensities were $\log_2$-transformed and then normalised to the intensity of IRS in each sample. Then, values were normalised to the mean of the insulin treated sample.

## Data and materials availability

RAW and MaxQuant processed data have been deposited in the PRIDE proteomeXchange repository and can be accessed at https://www.ebi.ac.uk/pride/archive/, using the accession PXD023441. The reactions, rate equations, differential equations, and parameter sets required to reproduce the models can be found in *Supplementary file 1*. The code for the modelling has been deposited to Github (*Nguyen Lab, 2021a*, copy archived at swh:1:rev:09b5d4f838bf60e790c10843fec901516845d7e2, *Nguyen Lab, 2021b*). Plasmids generated in this study will be made available upon request. Any further information and requests for resources should be directed to james.burchfield@sydney.edu.au or david.james@sydney.edu.au.

## Acknowledgements

The authors acknowledge the facilities, and the scientific and technical assistance, of the Australian Microscopy and Microanalysis Research Facility at the Charles Perkins Centre, University of Sydney. In particular, we would like to thank Neftali Florez-Rodriguez for technical assistance. This research was also facilitated by access to Sydney Mass Spectrometry, a core research facility at the University of Sydney. This work was funded by Australian Research Council (ARC) project grant number DP180103482 awarded to DEJ and JGB, and National Health and Medical Research Council (NHMRC) project grant number GNT1120201 awarded to DEJ. ALK was supported by a Research Training Program Scholarship, University of Sydney Postgraduate Merit Award and the Chen Family Research Scholarship. DMN and MKLW were supported by Australian Postgraduate Award Scholarships. SJH was supported by a fellowship from the University of Sydney (G197569). DJF was supported by a Medical Research Council Career Development Award (MR/S007091/1). LKN was supported by a Victorian Cancer Agency Mid-Career Research Fellowship (MCRF18026), an Investigator Initiated Research Scheme grant from National Breast Cancer Foundation (IIRS-20–094) and the Metcalf Venture Grants Scheme administered by Cancer Council Victoria, Australia. DEJ was supported by an NHMRC Senior Principal Research Fellowship.

## Additional information

### Competing interests

David E James: Reviewing editor, *eLife*. The other authors declare that no competing interests exist.

### Funding

| Funder | Grant reference number | Author |
|---|---|---|
| Australian Research Council | DP180103482 | David E James<br>James G Burchfield |
| National Health and Medical Research Council | GNT1120201 | David E James |
| University of Sydney | Research Training Program Scholarship | Alison L Kearney |
| University of Sydney | Australian Postgraduate Award Scholarships | Dougall M Norris<br>Martin Kin Lok Wong |
| University of Sydney | G197569 | Sean J Humphrey |
| Medical Research Council | MR/S007091/1 | Daniel J Fazakerley |
| Victorian Cancer Agency | MCRF18026 | Lan K Nguyen |
| National Breast Cancer Foun- | IIRS-20–094 | Lan K Nguyen |

dation

| Cancer Council Victoria | Metcalf Venture Grant | Lan K Nguyen |
| University of Sydney | Chen Family Research Scholarship | Alison L Kearney |
| University of Sydney | Postgraduate Merit Award | Alison L Kearney |

The funders had no role in study design, data collection and interpretation, or the decision to submit the work for publication.

### Author contributions

Alison L Kearney, Dougall M Norris, Conceptualization, Formal analysis, Investigation, Visualization, Methodology, Writing - original draft, Project administration, Writing - review and editing; Milad Ghomlaghi, Modelling; Martin Kin Lok Wong, Conceptualization, Formal analysis, Investigation, Methodology, Modelling; Sean J Humphrey, Formal analysis, Investigation, Methodology, Writing - review and editing; Luke Carroll, Formal analysis, Investigation, Methodology; Guang Yang, Thomas A Geddes, Investigation; Kristen C Cooke, Investigation, Project administration; Pengyi Yang, Formal analysis; Sungyoung Shin, Supervision, Modelling; Daniel J Fazakerley, Supervision, Funding acquisition, Writing - review and editing; Lan K Nguyen, Supervision, Writing - review and editing, Modelling; David E James, Conceptualization, Resources, Supervision, Funding acquisition, Writing - original draft, Writing - review and editing; James G Burchfield, Conceptualization, Resources, Formal analysis, Supervision, Funding acquisition, Investigation, Visualization, Methodology, Writing - original draft, Project administration, Writing - review and editing

### Author ORCIDs

Alison L Kearney https://orcid.org/0000-0002-5736-3393
Milad Ghomlaghi https://orcid.org/0000-0001-9047-1049
Lan K Nguyen https://orcid.org/0000-0003-4040-7705
David E James https://orcid.org/0000-0001-5946-5257
James G Burchfield https://orcid.org/0000-0002-6609-6151

### Decision letter and Author response
Decision letter https://doi.org/10.7554/eLife.66942.sa1
Author response https://doi.org/10.7554/eLife.66942.sa2

## Additional files

### Supplementary files
- Supplementary file 1. Reactions and rate equations for the PI3K/Akt pathway models.
- Supplementary file 2. Parameter identifiability analysis of models 4 to 9.
- Supplementary file 3. DNA sequence of plasmids and primers generated in this study.
- Transparent reporting form

### Data availability
RAW and MaxQuant processed data have been deposited in the PRIDE proteomeXchange repository and can be accessed at https://www.ebi.ac.uk/pride/archive/, with the accession PXD023441. The reactions, rate equations, differential equations and parameter sets required to reproduce the models can be found in the Supplementary File 1. The code for the modelling has been deposited to github and can be accessed at https://github.com/NguyenLab-IntegratedNetworkModeling/Akt-IRS-negative-feedback.git (copy archived at https://archive.softwareheritage.org/swh:1:rev:09b5d4f838bf60e790c10843fec901516845d7e2). Plasmids generated in this study will be made available upon request. Any further information and requests for resources should be directed to james.burchfield@sydney.edu.au or david.james@sydney.edu.au.

The following dataset was generated:

| Author(s) | Year | Dataset title | Dataset URL | Database and Identifier |
|-----------|------|---------------|-------------|-------------------------|
| Kearney AL, Humphrey SJ, Burchfield JG, James DE | 2021 | Akt phosphorylates insulin receptor substrate (IRS) to limit PI3K-mediated PI(3,4,5)P3 synthesis | https://www.ebi.ac.uk/pride/archive/projects/PXD023441 | PRIDE, PXD023441 |

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
