## [Decision Letter]

**Acceptance summary:**

This study demonstrates that AKT-mediated IRS1/2 protein phosphorylation provides a mechanism for negative feed-back regulation of insulin receptor signaling. Comprehensive analysis using computational modeling and experimental analysis provides convincing evidence to support the authors' conclusions.

**Decision letter after peer review:**

Thank you for submitting your article "Akt phosphorylates insulin receptor substrate (IRS) to limit PI3K-mediated PI(3,4,5)P3 synthesis" for consideration by *eLife*. Your article has been reviewed by 4 peer reviewers, one of whom is a member of our Board of Reviewing Editors, and the evaluation has been overseen by Jonathan Cooper as the Senior Editor. The reviewers have opted to remain anonymous.

Essential revisions:

1. Model results were obtained by computational simulation, with parameters determined from fitting to experimental data for certain nodes in the signaling network. Comparing the simulated results for the different alternative models permitted inferences about which assumed feedback mechanisms were more versus less consistent with the experimental data. This integrated framework is a very powerful approach for helping dissect the complex system being examined.

a) Can the authors provide information with respect to parameter identifiability and uncertainty? Without this information being described concretely in the text – and perhaps in appropriate figure panels – it is difficult for a reader to know what degree of confidence can be given to the comparisons among mechanisms.

b) When a quantitative assessment of "fit" is illustrated (e.g., Figure 2B), in terms of the objective function – which is an aggregate overall time as well as all variables – what are the associated effects on qualitative behavior of the dynamics? Given that a key point being investigated here is the interesting qualitative behavior of a feedback system, seeing how that relates to an aggregate numerical fit metric would be quite helpful.

c) When modeling the molecular validation experiments, how are the parameters altered by a given perturbation? For instance, are inhibitory perturbations assumed to be complete and exclusively precise (neither of which generally guaranteed experimentally)? A note on this point for each perturbation should be provided, again to buttress confidence in the model / experiment integration.

d) The code used for modeling should be made available.

2. An important open question that is not resolved by the study is the mechanism by which the IRS proteins interact with the plasma membrane prior to insulin stimulation. This is fundamental to the model proposed in the manuscript. However, in the model shown in Figure 7 only the interaction of the IRS proteins with the IR after insulin stimulation is shown and the receptor-independent interactions are not included. This issue should be noted. Any insight that can be provided into this outstanding question would strengthen the impact of the study.

3. The levels of plasma membrane associated IRS1 and IRS2 rapidly decrease almost immediately upon stimulation with insulin. However, PI3K and AKT activity are sustained over a longer time course. Do the IRS proteins stay associated with the HDM or LDM fractions for a longer period of time? This analysis is shown for the IR but not the IRS proteins.

4. Stimulation with EGF causes a decrease in PM associated IRS1 and IRS2. Is this AKT-dependent? Does EGF stimulation inhibit subsequent insulin-dependent PI3K activation?

5. The data shown in Figure 6 indicate that IRS1 and IRS2 S to A mutants increase interactions with the PM upon stimulation with insulin. Is this an increase in the interaction of these adaptors with the IR or is this a receptor-independent interaction as observed under basal conditions. In other words, is the phosphorylation of the IRS proteins disrupting receptor interactions or receptor-independent interactions at the PM?

6. The discussion could address the following topics:

a) The position and number of S/T phospho-sites mediate the feedback regulation should be discussed. How heterologous kinases co-opt this mechanism to mediate inflammatory or ER stress during metabolic stress should also be discussed.

b) In addition to negative feedback, the authors might address the possibility that separation of IRS from the PM/InsR might facilitate translocation of the IRS signaling complex to another cellular site, which could be important for signal propagation.

c) Can the authors discuss how the mechanism might provide a rationale for the evolution of IRS as an obligate intermediate between the InsR and PI3K.

d) The overshoot is most apparent at 1 nM and less obvious at 100 nM insulin. How does the concentration of the principle signaling components (insulin, IR, IRS) modulates the strength of the AKT-mediated modulation of downstream signaling?

e) The authors might mention that other sites modulate tyrosine phosphorylation of IRS1 (PMID: 24652289), and whether these other sites might have a similar mechanism as offered by this manuscript.

f) Additional discussion about the physiological role of the correction of acute AKT activation "overshoot" that is observed would also be helpful for communicating the overall significance of the study.

*Reviewer 1:*

This is an interesting study that examines negative feedback regulation of IRS signaling to PI-3K/AKT. The foundation for the analysis is the knowledge that there is a rapid overshoot of the recruitment of AKT to the plasma membrane at low insulin concentrations. A theoretical framework is presented and experimentally tested. The authors conclude that AKT-mediated phosphorylation of IRS proteins mediates rapid negative regulation by depleting the pool of plasma membrane-associated IRS proteins. Strong evidence is provided showing that two sites of AKT phosphorylation on IRS2 are important. It is suggested that IRS1 is regulated by a similar mechanism, but it is less clear which sites are of key importance because of the number of sites and potential redundancy (or functional cooperation). Overall, this is a strong study (with data that supports the authors conclusions) that advances our understanding of insulin signaling.

*Reviewer 2:*

This is an interesting and rigorous paper that provides new information on a mechanism of feedback signaling/regulation that modulates downstream insulin signaling activated through the PI3K→AKT signaling cascade. Feedback inhibition at the insulin receptor substrates is important to understand because it can contribute to the progression of type 2 diabetes during hyperinsulinemia owing to nutrient overload. The authors report rigorous analysis of PI3K→IRS→AKT association at the PM, and how AKT promotes dissociation of IRS from the PM. The dissociation provide a mechanism to explain results from many published report over the past 20 years. The methods and approach are very powerful and will be used in the future to investigate signaling at the PM. By fitting theoretical models to experimental results, the authors conclude that dissociation of IRS from the PM is mediated by S/T-phosphorylation of IRS. Direct measures of S/T-phosphorylation are limited to inferences from mutations and two MS identifications in IRS2. How S/T-phosphorylation promotes dissociation is not clear, and whether it involves specific interactions with the InsR or other PM components, or nonspecific electrostatic effects is not resolved.The conclusions are generally consistent with previous results that IRS tyrosine phosphorylation is inhibited by PI3K◊AKT mediated IRS S/T-phosphorylation (For example PMID: 24652289). Indeed, previous studies show that LIRKO mice show a loss of insulin stimulated IRS1 S/T phosphorylation in the liver but not in the muscle (PMID: 26846849), suggesting that insulin signaling itself is the major mediator of IRS S/T-phosphorylation; however, except for 2 sites in IRS2, the AKT sites involved in IRS1 were not resolved in this report.

This manuscript provides evidence to support and solidify the hypothesis that AKT-mediated S/T-phosphorylation of IRS drives the IRS-PI3K complex away from the PM/InsR to explain the feedback inhibition, including reduced PIP3 production and target recruitment. While evidence of IRS S/T-phosphorylation is generally indirect (or theoretically validated), this feedback mechanism and especially the role for AKT is consistent with previous results showing that AKT inhibitors along with inhibitors of other kinases in the cascade can have a positive effect upon the steady state IRS1 tyrosine phosphorylation. Increased Tyr phosphorylation would be expected for an IRS complex that lingers at the PM/InsR, and inclusion of such results might be helpful.

Although MS analysis of putative AKT phosphorylation sites in both IRS1 and IRS2 was difficult to achieve, mutation of six predicted AKT sites in IRS1 or 5 AKT sites in IRS2 enhanced PM association of IRS, AKT and PDPK1, which is consistent with increased IRS signaling (and presumably Tyr phosphorylation). A measure of Tyr phosphorylation in this case would be a worthwhile addition.

Mutation of AKT sites individually did not disrupt the AKT-mediated feedback inhibition, which was taken as evidence that multisite phosphorylation is key for feedback regulation, as previous suggested (PMID: 24652289).

Only in the case of IRS2 were combined mutations at two validated AKT sites (S306A and S577A) found to disrupt IRS2 dissociation and negative feedback through AKT. The interpretation of this result is reasonable to support the idea that multisite S/T-phosphorylation (rather than any single specific site) mediates dissociation of IRS from the PM/InsR; however, it seems surprising that two sites are so much more effective than one if the mechanism relies upon a nonspecific electrostatic repulsion; however, as a similar mechanism for IRS1 was not supported by the data, the authors might consider other AKT targets might be involved in the feedback mechanism at the PM.*Reviewer 3:*

The focus of this study is to examine the feedback regulation of the PI3K/AKT signaling network in response to insulin stimulation. Insulin signaling is under tight control and many feedback regulatory pathways control the intensity and longevity of these signals. The IRS proteins have been shown in many studies to play an important role in the feedback regulation of insulin signaling through their phosphorylation on multiple Ser/Thr residues that impact receptor interactions, effector interactions and degradation of the proteins. A number of different kinases have been implicated in this feedback, including AKT for IRS1. The novelty of the current study is that a specific role for AKT-dependent phosphorylation of IRS1 and IRS2 in an early, acute regulation of PI3K/AKT signaling is identified. Specifically, the rapid spike that occurs immediately after stimulation is reduced partially and then a steady state level of activity is observed, and the decline in activity is regulated by AKT-dependent phosphorylation. Specific phosphorylation sites on IRS2 are identified that mediate this acute decrease in IRS/PI3K/AKT signaling.Overall the study is nicely done and the data presented are clear and convincing. Computational modeling is validated by experimental conditions and a role for AKT in feedback regulation through the IRS proteins is demonstrated. The importance of this signaling pathway and its regulation for normal metabolic homeostasis and pathological conditions such as Type 2 diabetes and cancer makes the study significant and relevant for a broad audience.

An important open question that is not resolved by the study is the mechanism by which the IRS proteins interact with the plasma membrane prior to insulin stimulation. This is fundamental to the model proposed in the manuscript. However, in the model shown in Figure 7 only the interaction of the IRS proteins with the IR after insulin stimulation is shown and the receptor-independent interactions are not included.*Reviewer 4:*

I will restrict my comments to the mathematical modeling aspect of the study.The goal of the mathematical modeling aspect of the study was to put into explicit and quantitative terms key assumptions and hypotheses concerning the biochemical processes under consideration. Different, alternative models were formulated, and numerical results from each were simulated, in order to permit comparison of predictions arising from diverse postulates representing dynamic signaling pathway mechanisms, in particular feedback influences.

To the extent that I can assess from the manuscript, the modeling work appears to have been soundly performed.

---

## [Author Response]

Essential revisions:1. Model results were obtained by computational simulation, with parameters determined from fitting to experimental data for certain nodes in the signaling network. Comparing the simulated results for the different alternative models permitted inferences about which assumed feedback mechanisms were more versus less consistent with the experimental data. This integrated framework is a very powerful approach for helping dissect the complex system being examined.a) Can the authors provide information with respect to parameter identifiability and uncertainty? Without this information being described concretely in the text – and perhaps in appropriate figure panels – it is difficult for a reader to know what degree of confidence can be given to the comparisons among mechanisms.

Due to the technical nature of the question and response we provide here both a brief overview (designed to be accessible to all) and a more detailed response specifically intended for the Reviewers.

Overview: The Reviewers’ question pertains to whether or not a given parameter in our models can be assigned a degree of statistical confidence. If it can – it is considered identifiable and if not then it is non identifiable. An identifiable model is one in which all of the parameters that define the model are identifiable. We have now performed this analysis (Supplementary File 2, main text) and show that our models contain a mix of identifiable and non-identifiable models. As such the overall model is considered non-identifiable. This is not unexpected as the majority of large models are non-identifiable, because identifiability is a tradeoff between model complexity and the amount of biological data available to train the model. So our models could be made identifiable by simplifying them at the cost of biological detail. Rather, our goal was to explore/capture the biological detail in order to narrow down the mechanism using an iterative approach of modeling, prediction and experimental validation. Retraining the models with the data from the experiments that were used to test and validate the model predictions (Figures 2D, E-G,I; 3; 4B,C, main text) is expected to improve the identifiability.

Detailed Response: We agree that model identifiability analysis provides useful information about the identifiability of the model parameters during model calibration, and should be included. Thus in response to the Reviewers’ comment, we have now undertaken identifiability analysis for all the tested models (models 4 to 9, Figure 2A, main text) using a well-established method based on profile likelihood (Maiwald et al., 2016; Rateitschak et al., 2012; Raue et al., 2009). This method is able to detect both structural and practical non-identifiable parameters by calculating confidence intervals defined by a threshold in the profile likelihoods, as detailed in this method paper (Rateitschak et al., 2012). A parameter is considered to be identifiable if its confidence interval of its estimate is finite (Rateitschak et al., 2012). Another important advantage of this approach is that it is computationally efficient and thus suitable for medium-to-large (non-minimal) models such as those considered in this paper (Rateitschak et al., 2012).

Briefly, for the parameter estimation we used the following weighted sum of squared residuals:χ2(θ)=∑j=1M∑i=1N(yj,iD−yj(θ)σj,i)2where yj,iD are the experimental data, yj(θ) are model simulations at time points that the experimental data are measured and σj,i is measurement error of the experimental data. For normally distributed observational noise, this function corresponds to the maximum likelihood estimate (MLE) of θ.

The profile likelihood of a parameter θ is given by (Maiwald et al., 2016; Rateitschak et al., 2012; Raue et al., 2009).

χPL2(θi)=minθj≠iχ2(θ),

which represents a function in θi of least increase in the residual sum of squares χ (θ).

Profile likelihood-based confidence interval (CI) can be derived via

CI(θ)={θ|χPL2(θ)−χPL2(θ^)<Δα},

where Δα=χ2(α,df) is the threshold, α is a confidence level (the α quantile of the χ^2^-distribution),df is the degree of freedom (df=1 for pointwise confidence interval and df=#ofparameters for simultaneous confidence intervals, respectively). θ^ denotes the best-fitted parameter set.

The identifiability analysis results for models 4-9 are now presented in the manuscript (new Supplementary File 2, main text). In each plot the black dashed lines depict pointwise confidence levels of . The results show for each parameter in a model whether it is structurally (indicated by a flat curve in both directions (Rateitschak et al., 2012)) or practically (indicated by a flat curve only in one direction (Rateitschak et al., 2012)) non-identifiable. Overall, these results demonstrate that the models are non-identifiable, meaning each model has at least one non-identifiable parameter.

As most large ODE-based models in systems biology are unidentifiable to some extent, the results here were expected given the detailed scope of our models, which were designed to capture the important biological mechanisms within the insulin signalling network, including multiple feedback/feedforward loops and crosstalk. Generally there is a trade-off between model identifiability and the level of biological detail, so whilst the models in this study could be made more identifiable through model abstraction, this would be at the cost of scarifying specific biological details and may weaken the models’ explanatory and predictive power.

While the new identifiability analysis results provide additional information about our candidate models, they alone do not allow us to discriminate between the models. Instead, in this study we have adopted a step-wise approach that progressively rules out less-confident models through fitting quality to calibrated datasets and then independent validation with new sets of data. Specifically, we first calibrated the proposed hypothetical models (models 4-9) using experimental data and then quantitatively (and qualitatively, see response to point b below) evaluated the fitting quality of the models, assuming better-fitted models are more likely to reflect ‘true’ biological reality. As a result, we narrowed down to models 7-9 as the better fitted models given their lower discrepancies between simulation and experimental data and considered them for further investigations. We next used a series of simulations/predictions paired with biological testing in order to discriminate between these models (Figures 2C-D,H-I, 4A,B, main text). Experimental data was only consistent with Model 9 and this mechanism was further validated experimentally (Figures 5-7, main text).

Following the Reviewers’ comment, we have provided all the additional identifiability analysis in the revised manuscript (Supplementary File 2, main text). In addition, we have now included a brief summary of this analysis in the main text of the revised manuscript on page 7: “Identifiability analysis (Maiwald et al., 2016; Rateitschak et al., 2012; Raue et al., 2009) demonstrated that all models were similarly unidentifiable, with at least one parameter not identifiable in each model (Supplementary File 2), which is not surprising given the size of our models (see Materials and methods for detailed description).” and a detailed description in the Materials and methods section on page 30.

b) When a quantitative assessment of "fit" is illustrated (e.g., Figure 2B), in terms of the objective function – which is an aggregate overall time as well as all variables – what are the associated effects on qualitative behavior of the dynamics? Given that a key point being investigated here is the interesting qualitative behavior of a feedback system, seeing how that relates to an aggregate numerical fit metric would be quite helpful.

Following the Reviewers’ comment, we have further analyzed the qualitative behaviors of the models and compared their simulated dynamics using the best-fitted parameter set to the corresponding experimental data. Consistent with the poor quantitative fitting indicated by the numerical objective function (Figure 2B, main text), models 4-6 also display poor qualitative fitting for key model species. For example, as shown in Author response image 1, model 4 failed to reproduce a strong overshoot of PM Akt dynamics in response to 100 nM insulin (red boxes, Author response image 1). Model 4 also showed a significant qualitative difference in the dynamics of pAkt S474 between simulation and experimental data. Moreover, model 5 displayed a significant delay in pAkt T309 in response to 1 nM insulin, which is inconsistent with experimental data. Model 6 did not qualitatively reproduce the overshoot of PM Akt following 1 nM insulin stimulation, and in this model pAkt T309 peak showed a considerable delay (Author response image 1). On the other hand, models 7-9 showed better performance based on the numerical objective function values (Figure 2B, main text) and the simulated dynamics showed improved qualitative consistency with the experimental data, as shown in Figure 2—figure supplement 1D, E, F.

Overall, we observed a strong consistency between quantitative assessment of the model fitting and qualitative agreement with experimental data.

**Author response image 1. sa2fig1:** Models’ simulations of Akt recruitment, T309 and S474 phosphorylation, and PRAS40 phosphorylation in response to 1 and 100 nM insulin (simulation; dotted lines). This was overlaid with the experimentally observed kinetics for each outcome (experiment; solid lines, mean ± S.E.M). Red boxes highlight inconsistencies between model’s simulations and experimental data.

In the revised manuscript, we have now included a new paragraph on page 7 to incorporate this new analysis: “Quantitative assessment of model fitting based on the objective function revealed that Akt to IRS/PI3K (model 9) best matched the experimental data, followed by mTORC2 to PTEN (model 7), and then Akt to PTEN (model 8) (Figure 2B). […] Overall, since models 7-9 displayed superior quantitative and qualitative consistency with experimental data we focused on interrogating these models hereafter.”

c) When modeling the molecular validation experiments, how are the parameters altered by a given perturbation? For instance, are inhibitory perturbations assumed to be complete and exclusively precise (neither of which generally guaranteed experimentally)? A note on this point for each perturbation should be provided, again to buttress confidence in the model / experiment integration.

We agree with the Reviewers’ comment – inhibitory perturbations, especially small molecules, are likely not complete or entirely precise. However, for the purposes of modelling signalling outcomes, inhibitory perturbations were assumed to be both complete and precise. In this case, we believe it was appropriate to make this assumption because we have extensive experience with the small molecules (MK2206, GDC0068, rapamycin) and genetic (W80A system) approaches we have used in the cell type being studied. In general, we observe very high degrees of signalling inhibition when assessing insulin signaling responses by different methods that include monitoring protein translocation by microcopy (e.g. TIRFM) and protein phosphorylation by mass spectrometry-based proteomics and Western blot. We provide a note for each perturbation below:

Chemical-genetics: For loss of T309 and S474 phosphorylation through mutation we utilised the W80A system. Since prevention of phosphorylation is genetically encoded they are by definition complete and specific (Kearney et al., 2019).

Akt inhibition: MK2206 is an allosteric inhibitor of Akt with high specificity for Akt (Wiechmann et al., 2021) and in our context shows complete inhibition of Akt recruitment and thus kinase activity at 10 µM (Kearney et al., 2019). Similarly, GDC0068 results in near complete inhibition of Akt kinase activity at 10 µM (Figure 2—figure supplement 2, main text).

mTORC1 inhibition: Rapamycin has been extensively used and validated previously to ablate mTORC1-mediated phosphorylation of S6K (Humphrey et al., 2013; Yang et al., 2015) and in our context we have shown that 20 nM Rapamycin lowered mTORC1-mediated S6K phosphorylation to levels observed in unstimulated cells (Figure 1H, main text).

PDPK1 inhibition: GSK2334470 decreased phosphorylation of Akt at T309 (as expected for a PDPK1 inhibitor) and Akt-mediated phosphorylation of AS160 at 10 µM to levels observed in unstimulated cells (Figure 3—figure supplement 1, main tex**t**).

To account for the fact that inhibitory perturbations are often not precise (have off-target effects), throughout our study we validated the modelling predictions using multiple corroborating experimental validations. For example, the modelling predicted that loss of negative feedback (through Akt inhibition) would result in the hyper recruitment of Akt to the plasma membrane (Figure 2C, main text). We tested this experimentally using GDC0068 (Figure 2D, main text), as well as three Akt mutants using the W80A system (W80A-T309A, W80A-T309A-S474A, W80A-K181A) (Figure 2F,I, main text).

Following the Reviewers’ comment, we have now provided all the relevant *in silico* perturbation conditions in the figure legends of the revised manuscript accordingly as follows:

In Figure 1F, G: “Model prediction of the effect of mTORC1 inhibition on Akt recruitment in response to 1 nM insulin. The value of parameter Ki2 was set to null to model complete inhibition of mTORC1 catalytic activity.”

In Figure 2H: “Model predictions of the effect of losing Akt T309 or Akt S474 phosphorylation on Akt recruitment in response to 1 and 100 nM insulin. The parameter values of Kf6 and Kf7 was set to null to represent loss of Akt T309 and Akt S474 phosphorylation, respectively.”

In Figure 2C: “Predictions from models 7-9 on the effect of Akt inhibition on Akt recruitment in response to 1 nM insulin. The parameter values of Ki2a, Ki2b, Kf11a, and Kf11b were set to null in order to model Akt inhibition by GDC0068.”

In Figure 4A: “Model predictions detailing the effect of PI3K inhibition following insulin stimulation, on the plasma membrane dissociation of Akt with and without Akt inhibition. The parameter value of Kf3 was set to null to simulate PI3K inhibition by wortmannin.”

d) The code used for modeling should be made available.

The code for the modelling has been deposited to github and can be accessed at https://github.com/NguyenLab-IntegratedNetworkModeling/Akt-IRS-negative-feedback.git. The manuscript ‘Data and Materials Availability’ statement on page 52 has been modified accordingly.

2. An important open question that is not resolved by the study is the mechanism by which the IRS proteins interact with the plasma membrane prior to insulin stimulation. This is fundamental to the model proposed in the manuscript. However, in the model shown in Figure 7 only the interaction of the IRS proteins with the IR after insulin stimulation is shown and the receptor-independent interactions are not included. This issue should be noted. Any insight that can be provided into this outstanding question would strengthen the impact of the study.

We agree that this is an important question and that it should be noted. Interaction of IRS1/2 with the insulin receptor is not observed in the basal state, and is only detected following insulin stimulation (Figure 5J). This led us to hypothesise that the IRS:plasma membrane (PM) interaction prior to insulin stimulation is independent of receptor interactions, and most likely mediated by the pleckstrin homology (PH) domain interacting with phospholipids in the PM such as PI(4,5)P2 (Dhe-Paganon et al., 1999). In order to test this hypothesis, we utilised information provided by previous IRS1 structural studies (Dhe-Paganon et al., 1999) to delete the PH domain from IRS1-eGFP (DelPH IRS1-eGFP). We focused on IRS1, as IRS2 lacks this structural information. Deletion of the IRS1 PH domain reduced the TIRF/EPI ratio in the basal state (new Figure 5F main text), consistent with loss of PM-associated IRS1.

Further, we did not detect a rapid decrease in the TIRF signal of DelPH-IRS1-eGFP following insulin stimulation, like we do for WT IRS1 (Author response image 2). This is consistent with cytosolic localisation. These data suggest that the IRS1 PH domain is largely responsible for its localisation at the PM prior to insulin stimulation. We note that these data do not preclude a PH domain dependent interaction with a PM-localised protein/complex.

**Author response image 2. sa2fig2:** 3T3-L1 adipocytes were electroporated with WT (left) or DelPH (right) IRS1-eGFP and stimulated with 1 nM Insulin. Recruitment was assessed by TIRFM (three independent experiments, data expressed as mean ± S.E.M, PM; plasma membrane).

We have also modified the model in new Figure 7, main text to include these receptor-independent interactions in the basal state, and have indicated in the figure that these are PH domain-mediated.

We have modified the text of the manuscript to describe these data on page 16 – “We attribute the PM localisation of IRS1/2 prior to insulin stimulation to its pleckstrin homology (PH) domain, as deletion of the IRS1 PH domain (DelPH IRS1-eGFP) decreased its abundance at the PM in unstimulated cells (Figure 5F). […] Together, these data indicate that IRS1/2 is PM-localised in unstimulated cells via a PH domain-dependent interaction, and upon insulin stimulation Akt induces the translocation of IRS1/2 from the PM to the cytosol.”, and on page 22 - “We propose a model where in unstimulated cells, a pool of IRS is localised to the PM via a PH domain-dependent interaction. […] Ultimately, these events limit PM-associated PI3K and PIP3 synthesis (Figure 7).”

3. The levels of plasma membrane associated IRS1 and IRS2 rapidly decrease almost immediately upon stimulation with insulin. However, PI3K and AKT activity are sustained over a longer time course. Do the IRS proteins stay associated with the HDM or LDM fractions for a longer period of time? This analysis is shown for the IR but not the IRS proteins.

We now include data probing the abundance of IRS1 in the HDM and LDM fractions following 10 min insulin stimulation in the presence or absence of Akt inhibitors (GDC0068 and MK2206) (Figure 5E). In all conditions there is no change in the abundance of IRS in either of these fractions. Rather, as originally shown in Figure 5E, IRS moves from the plasma membrane (PM) to the cytosol fraction, and stays in the cytosol for a longer period of time (10 min). Our hypothesis is that following insulin stimulation IRS loss from the PM associated pool *limits* its association with the IR, however this IR:IRS interaction still occurs to some extent, and this sustains PI3K and Akt activity.

We have modified the text of the manuscript to describe these data on Page 16 – “Consistent with these observations, subcellular fractionation also demonstrated an Akt-dependent decrease in the abundance of endogenous IRS1 at the PM, and extended these findings to reveal that IRS1 moves from the PM to the cytosol upon insulin stimulation, rather than to internal membranes (high density microsome (HDM)/low density microsome (LDM) fractions) (Figure 5E).”

4. Stimulation with EGF causes a decrease in PM associated IRS1 and IRS2. Is this AKT-dependent? Does EGF stimulation inhibit subsequent insulin-dependent PI3K activation?

To test whether the decrease in PM associated IRS1/2 upon EGF stimulation is Akt-dependent, we performed TIRFM in adipocytes expressing IRS1-eGFP or IRS2-eGFP in the presence or absence of Akt inhibitors (GDC0068 and MK2206). These new data show that the decrease in PM-associated IRS1/2 with EGF is only partially Akt-dependent (Author response image 3). EGF decreased PM IRS1 in control cells, but rapidly increased in the abundance of IRS1/2 abundance at the PM in the presence of Akt inhibitors. This was quickly followed by a return to unstimulated levels (Author response image 3).

These data demonstrate that EGF stimulation causes an Akt-dependent decrease in PM associated IRS1/2. However, these data also revealed greater complexity in the regulation of PM IRS1/2 in response to EGF than we observed in response to insulin, and that other Akt-independent mechanisms regulate IRS localisation in response to EGF. For example, a variety of other kinases phosphorylate IRS such as JNK, ERK1/2, PKCs, S6K and mTORC1 (Copps and White, 2012). These kinases could regulate IRS1/2 localisation through phosphorylation and drive a similar mechanism to that described in this study. For example, EGF activates ERK1/2 more potently than insulin (van den Berghe et al., 1994), and so perhaps activation of ERK1/2 is contributing to these changes in IRS1/2 localisation with EGF. This is intriguing and suggests that IRS could be a central node that integrates information about the activation status of various kinases/pathways and as such serves as a key site of pathway crosstalk. Despite this, our view is that these new data (Author response image 3) would distract from the main message of the manuscript and would be best dissected in a separate study.

**Author response image 3. sa2fig3:** 3T3-L1 adipocytes were electroporated with IRS1-eGFP or IRS2-eGFP. Cells were stimulated with DMSO, 10 µM MK2206 or 10 µM GDC0068 5 min prior to 100 ng/mL EGF (epidermal growth factor) and recruitment assessed by TIRFM (two independent experiments, data expressed as mean ± S.E.M, PM; plasma membrane).

The control data only (Author response image 3) was used in the original manuscript to suggest that Akt facilitates the dissociation of IRS from the PM independently of the insulin receptor. As we now know this process is only partially Akt-dependent (Author response image 3), we have replaced these data with a more convincing experiment (new Figure 5I, main text), where we have deleted the phosphotyrosine binding (PTB) domain of IRS1 (which is responsible for IRS binding to the insulin receptor) and still observed removal of IRS1 from the PM upon insulin stimulation. This suggests that Akt directly facilitates the removal of IRS from the PM independently of the insulin receptor, in a more convincing manner than the original EGF experiment.

Based on our working model (Figure 7, main text), it makes sense that EGF would reduce the availability of IRS at the PM and therefore inhibit subsequent insulin-dependent PI3K activation. We have in fact observed a similar phenotype with insulin. Cells pre-stimulated with insulin (low dose – 1 nM) demonstrate a decreased rate of Akt recruitment in comparison to naive cells following stimulation with a higher dose (100 nM) (Author response image 4). However, we think these experiments are perhaps too simplistic – the effect of EGF (or insulin) stimulation on subsequent insulin-stimulated PI3K activation would likely extend past IRS localisation, particularly as both pathways share a lot of machinery. For example, EGF (or insulin) may promote insulin receptor activation (and thus PI3K activation) through inhibition of the insulin receptor phosphatase PTP1B (Ravichandran et al., 2001), or may engage a variety of other described positive and negative feedbacks (Brummer et al., 2008; Rodrigues et al., 2000) which may affect the propensity of PI3K to be further activated by insulin. Thus, we believe these data would not improve the manuscript. Furthermore, given we have now removed the EGF data for reasons explained above, exploring whether EGF stimulation inhibits subsequent insulin-dependent PI3K activation would not make logical sense in the manuscript.

**Author response image 4. sa2fig4:** 3T3-L1 adipocytes were electroporated with TagRFP-T-Akt2. Cells were stimulated with 100 nM insulin with or without a 1 nM insulin pre-stimulation. Recruitment was assessed by TIRFM (data expressed as mean ± S.E.M, PM; plasma membrane).

5. The data shown in Figure 6 indicate that IRS1 and IRS2 S to A mutants increase interactions with the PM upon stimulation with insulin. Is this an increase in the interaction of these adaptors with the IR or is this a receptor-independent interaction as observed under basal conditions. In other words, is the phosphorylation of the IRS proteins disrupting receptor interactions or receptor-independent interactions at the PM?

We agree that this is an important detail to include in the manuscript. The phosphotyrosine binding (PTB) domain of IRS is responsible for its interaction with the insulin receptor (Eck et al., 1996). To test whether the phosphorylation of the IRS proteins disrupt receptor interactions or receptor-independent interactions at the PM, we utilised information provided by previous IRS1 structural studies (Dhe-Paganon et al., 1999) to delete the PTB domain from IRS1-eGFP (DelPTB IRS1-eGFP). IRS2 lacks this structural information, and so we focused on IRS1. DelPTB IRS1-eGFP was still depleted from the PM following insulin stimulation (New Figure 5I main text). This suggests that Akt-dependent phosphorylation of IRS proteins disrupts receptor-independent interactions at the PM. This limits the size of the IRS pool available to interact with the insulin receptor (Figure 5J). So, the S to A mutants described in the study lead to an increased pool available for interaction, which in turn leads to an increased interaction with the insulin receptor, and ultimately increased PIP3.

As pointed out by the Reviewers, the abundance of IRS1/2 6/5P increases at the PM with insulin (Figure 6C-D). Here, these mutants have been overexpressed and endogenous IRS1/2 remains. Given these new data (New Figure 5I main text) we believe this increase is likely a result of Akt removing endogenous IRS from the PM through disrupting receptor-independent interactions and making ‘more room’ for IRS1/2 6/5P (which Akt can’t phosphorylate) at the PM. However, in a system with no mutant overexpression, loss of Akt-mediated IRS phosphorylation likely causes no change in IRS localisation upon insulin stimulation, as seen with Akt inhibitors (Figure 5A-B).

We have modified the text of the manuscript to describe these data on Page 16 – “Furthermore, deletion of the phosphotyrosine binding (PTB) domain of IRS1 (DelPTB IRS1-eGFP), which is responsible for the interaction of IRS1 with the IR (Eck et al., 1996), did not impact IRS1 removal from the PM upon insulin stimulation (Figure 5I). These data suggest Akt removes IRS from the PM via disruption of insulin receptor-independent interactions.”

6. The discussion could address the following topics:a) The position and number of S/T phospho-sites mediate the feedback regulation should be discussed. How heterologous kinases co-opt this mechanism to mediate inflammatory or ER stress during metabolic stress should also be discussed.

We have modified the text of the manuscript to discuss this on Page 23 – “Here, we have shown that the phosphorylation of two IRS2 serine residues by Akt synergise to induce IRS2 translocation to the cytosol (Figure 6G). […] It is important to note, however, that modulation of IRS serine/threonine phosphorylation is likely not the mechanism of insulin resistance (Fazakerley et al., 2019; Hoehn et al., 2008).”

b) In addition to negative feedback, the authors might address the possibility that separation of IRS from the PM/InsR might facilitate translocation of the IRS signaling complex to another cellular site, which could be important for signal propagation.

This is an interesting point. We cannot exclude the possibility that the feedback onto IRS from Akt that we have discovered serves an additional function alongside dampening signal transduction. We have addressed this in the discussion on Page 23 – “In addition to negative feedback, it is possible that this movement of IRS to the cytosol might facilitate translocation of the IRS signaling complex to another site, which could be important for signal propagation.”

c) Can the authors discuss how the mechanism might provide a rationale for the evolution of IRS as an obligate intermediate between the InsR and PI3K.

We have added text to the discussion on Page 24 – “We envisage that the described feedback mechanism may have co-evolved with IRS proteins as an obligate intermediate between the IR and PI3K to enhance the signaling capacity of the IR in several ways. […] Finally, as IRS1/2 can be phosphorylated by a range of other kinases (Copps and White, 2012), it is possible that the described mechanism transcends to other kinases and enables cross talk between different growth factor pathways.”

d) The overshoot is most apparent at 1 nM and less obvious at 100 nM insulin. How does the concentration of the principle signaling components (insulin, IR, IRS) modulates the strength of the AKT-mediated modulation of downstream signaling?

As the Reviewers points out, the overshoot in Akt recruitment is most apparent at 1 nM insulin and less so at 100 nM insulin. This suggests that between insulin doses there are differences in the nature of the engagement of the feedback mechanism described, and/or the engagement of other mechanisms which may influence the recruitment behaviour of Akt. We have included a potential explanation for describing the difference in Akt recruitment between 1 and 100 nM insulin on Page 5 – “In response to 100 nM insulin, the maximum recruitment of Akt was 4-fold higher than 1 nM insulin, with an initial overshoot followed by a secondary increase (Figure 1B), which may reflect the engagement of a positive feedback signal at this dose.”

Commenting further on this question would require comprehensive computational modelling, which we think is best reserved for a separate study.

e) The authors might mention that other sites modulate tyrosine phosphorylation of IRS1 (PMID: 24652289), and whether these other sites might have a similar mechanism as offered by this manuscript.

We have modified the text of the manuscript to discuss this on Page 23 – “Here, we have shown that the phosphorylation of two IRS2 serine residues by Akt synergise to induce IRS2 translocation to the cytosol (Figure 6G). […] It is important to note, however, that modulation of IRS serine/threonine phosphorylation is likely not the mechanism of insulin resistance (Fazakerley et al., 2019; Hoehn et al., 2008).”

f) Additional discussion about the physiological role of the correction of acute AKT activation "overshoot" that is observed would also be helpful for communicating the overall significance of the study.

The meaning of the phrase ‘the correction of acute AKT activation’ was not clear to us. We interpret this to mean that we should discuss the physiological role of the feedback (that generates the overshoot) in greater detail. To this end we have added text to the discussion on Page 24 – “We envisage that the described feedback mechanism may have co-evolved with IRS proteins as an obligate intermediate between the IR and PI3K to enhance the signaling capacity of the IR in several ways. […] Finally, as IRS1/2 can be phosphorylated by a range of other kinases (Copps and White, 2012), it is possible that the described mechanism transcends to other kinases and enables cross talk between different growth factor pathways.”

References:

Brummer T, Larance M, Herrera Abreu MT, Lyons RJ, Timpson P, Emmerich CH, Fleuren EDG, Lehrbach GM, Schramek D, Guilhaus M, James DE, Daly RJ. 2008. Phosphorylation-dependent binding of 14-3-3 terminates signalling by the Gab2 docking protein. EMBO J 27:2305–2316.

Copps KD, White MF. 2012. Regulation of insulin sensitivity by serine/threonine phosphorylation of insulin receptor substrate proteins IRS1 and IRS2. Diabetologia 55:2565–2582.

Dhe-Paganon S, Ottinger EA, Nolte RT, Eck MJ, Shoelson SE. 1999. Crystal structure of the pleckstrin homology-phosphotyrosine binding (PH-PTB) targeting region of insulin receptor substrate 1. Proc Natl Acad Sci U S A 96:8378–8383.

Eck MJ, Dhe-Paganon S, Trüb T, Nolte RT, Shoelson SE. 1996. Structure of the IRS^-1^ PTB domain bound to the juxtamembrane region of the insulin receptor. Cell 85:695–705.

Fazakerley DJ, Krycer JR, Kearney AL, Hocking SL, James DE. 2019. Muscle and adipose tissue insulin resistance: malady without mechanism? J Lipid Res 60:1720–1732.

Hançer NJ, Qiu W, Cherella C, Li Y, Copps KD, White MF. 2014. Insulin and metabolic stress stimulate multisite serine/threonine phosphorylation of insulin receptor substrate 1 and inhibit tyrosine phosphorylation. J Biol Chem 289:12467–12484.

Hoehn KL, Hohnen-Behrens C, Cederberg A, Wu LE, Turner N, Yuasa T, Ebina Y, James DE. 2008. IRS1-independent defects define major nodes of insulin resistance. Cell Metab 7:421–433.

Hornbeck PV, Zhang B, Murray B, Kornhauser JM, Latham V, Skrzypek E. 2015. PhosphoSitePlus, 2014: mutations, PTMs and recalibrations. Nucleic Acids Res 43:D512–20.

Humphrey SJ, Yang G, Yang P, Fazakerley DJ, Stöckli J, Yang JY, James DE. 2013. Dynamic adipocyte phosphoproteome reveals that Akt directly regulates mTORC2. Cell Metab 17:1009–1020.

Kearney AL, Cooke KC, Norris DM, Zadoorian A, Krycer JR, Fazakerley DJ, Burchfield JG, James DE. 2019. Serine 474 phosphorylation is essential for maximal Akt2 kinase activity in adipocytes. J Biol Chem 294:16729–16739.

Maiwald T, Hass H, Steiert B, Vanlier J, Engesser R, Raue A, Kipkeew F, Bock HH, Kaschek D, Kreutz C, Timmer J. 2016. Driving the Model to Its Limit: Profile Likelihood Based Model Reduction. PLoS One 11:e0162366.

Rateitschak K, Winter F, Lange F, Jaster R, Wolkenhauer O. 2012. Parameter identifiability and sensitivity analysis predict targets for enhancement of STAT1 activity in pancreatic cancer and stellate cells. PLoS Comput Biol 8:e1002815.

Raue A, Kreutz C, Maiwald T, Bachmann J, Schilling M, Klingmüller U, Timmer J. 2009. Structural and practical identifiability analysis of partially observed dynamical models by exploiting the profile likelihood. Bioinformatics 25:1923–1929.

Ravichandran LV, Chen H, Li Y, Quon MJ. 2001. Phosphorylation of PTP1B at Ser(50) by Akt impairs its ability to dephosphorylate the insulin receptor. Mol Endocrinol 15:1768–1780.

Rodrigues GA, Falasca M, Zhang Z, Ong SH, Schlessinger J. 2000. A novel positive feedback loop mediated by the docking protein Gab1 and phosphatidylinositol 3-kinase in epidermal growth factor receptor signaling. Mol Cell Biol 20:1448–1459.

van den Berghe N, Ouwens DM, Maassen JA, van Mackelenbergh MG, Sips HC, Krans HM. 1994. Activation of the Ras/mitogen-activated protein kinase signaling pathway alone is not sufficient to induce glucose uptake in 3T3-L1 adipocytes. Mol Cell Biol 14:2372–2377.

Wiechmann S, Ruprecht B, Siekmann T, Zheng R, Frejno M, Kunold E, Bajaj T, Zolg DP, Sieber SA, Gassen NC, Kuster B. 2021. Chemical Phosphoproteomics Sheds New Light on the Targets and Modes of Action of AKT Inhibitors. ACS Chem Biol. doi:10.1021/acschembio.0c00872

Yang G, Murashige DS, Humphrey SJ, James DE. 2015. A Positive Feedback Loop between Akt and mTORC2 via SIN1 Phosphorylation. Cell Rep 12:937–943.